# ON THE COMPLEXITY OF NONSMOOTH AUTOMATIC DIFFERENTIATION

**Jérôme Bolte[1,2], Ryan Boustany[1,2,4], Edouard Pauwels [2,3] & Béatrice Pesquet-Popescu [4]**
[1] Toulouse School of Economics
[2] Université de Toulouse
[3] IRIT, CNRS. Institut Universitaire de France (IUF).
[4] Thales LAS France
{jerome.bolte, ryan.boustany}@ut-capitole.fr, edouard.pauwels@irit.fr,
beatrice.pesquet@thalesgroup.com

## ABSTRACT

Using the notion of conservative gradient, we provide a simple model to estimate the computational costs of the backward and forward modes of algorithmic differentiation for a wide class of nonsmooth programs. The overhead complexity of the backward mode turns out to be independent of the dimension when using programs with locally Lipschitz semi-algebraic or definable elementary functions. This considerably extends Baur-Strassen's smooth cheap gradient principle. We illustrate our results by establishing fast backpropagation results of conservative gradients through feedforward neural networks with standard activation and loss functions. Nonsmooth backpropagation's cheapness contrasts with concurrent forward approaches, which have, to this day, dimensional-dependent worst-case overhead estimates. We provide further results suggesting the superiority of backward propagation of conservative gradients. Indeed, we relate the complexity of computing a large number of directional derivatives to that of matrix multiplication, and we show that finding two subgradients in the Clarke subdifferential of a function is an NP-hard problem.

## 1 INTRODUCTION

**Automatic evaluation of derivatives:** Algorithmic differentiation (AD) appeared around 60 years ago (Beda et al. (1959); Wengert (1964)), and has been since then constantly developed and used in many contexts, see Griewank et al. (1989); Griewank and Walther (2008) for a thorough discussion. Today, it is at the core of modern learning architectures (Rumelhart et al., 1986; LeCun et al., 2015; Baydin et al., 2018), to the point that training a neural network (NN) is ultimately a way to combine the outputs of AD. There are many practical and theoretical developments available nowadays: flexible and efficient numerical libraries (Abadi et al., 2016; Paszke et al., 2019; Bradbury et al., 2018), an implicit differentiation theory (Griewank and Faure, 2003; Griewank and Walther, 2008) and its extensions (Agrawal et al., 2019; Bai et al., 2019; Bolte et al., 2021; Blondel et al., 2021), the adjoint method (Farrell et al., 2013; Pearlmutter, 1995; Plessix, 2006) with application to neural ODEs (Chen et al., 2018), "piggyback" style differentiation of optimization algorithms (Griewank and Faure, 2003; Mehmood and Ochs, 2020; Bertrand et al., 2020; Lorraine et al., 2020), or differentiation of conjugate gradient algorithms (Gratton et al., 2014).

Backward algorithmic differentiation, or backpropagation, plays a particular role when smooth optimization tasks are at stake, as it evaluates the gradient of a function with a cost proportional to that of function evaluations, independently of dimension. This property, called *the cheap gradient principle* (Wolfe, 1982; Griewank and Walther, 2008), is at the root of the machine learning libraries revolution. According to the key complexity theory version of this result due to Baur and Strassen (1983), arithmetic complexity of the evaluation of the derivative of a rational function is at most 5 times the complexity of function evaluation. Extensions exist for smooth differentiable functions Baur and Strassen (1983); Griewank and Walther (2008) but standard computational practice of AD consists of little known about the nonsmooth case.

The objective of this paper is precisely to present a simple, general, nonsmooth cheap conservative principle and to explore other complexity results for evaluating nonsmooth derivatives. This extends the cheap gradient principle of smooth AD to the path differentiable world Bolte and Pauwels (2020b) which includes semi-algebraic and more generally definable functions Coste (2000a;b), a class that contains the vast majority of machine learning programs used in practice, see for example Bolte and Pauwels (2020b).

**Nonsmooth AD & computational complexity:** Sorting values, pooling data, thresholding functions, or determining closest points are some of the most essential numerical decision operations. They are ubiquitous in machine learning and modern optimization. All of them are nonsmooth, and most of them have a very desirable feature: they are cheap to compute, much cheaper than smoothed surrogates. For instance, the famous ReLU activation in deep learning, whose role is to threshold to zero negative values to allow for the inactivity of neurons, requires only one bit of encoding in theory. On the other hand, other nonlinear activations potentially require auxiliary algorithms for their evaluation, incurring a higher computational cost. This simplicity of use also comes with the issue of finding an adequate way of training models and, thus differentiating objects.

The standard computational practice of AD consists in applying differential calculus rules directly to nonsmooth objects, replacing gradients by surrogates, typically Clarke subgradients. This is how AD is performed within TensorFlow, PyTorch or Jax. This approach has shown tremendous success (LeCun et al., 2015) and has been massively used for the last 10 years. Yet, despite this empirical success, Barton et al. claimed in Barton et al. (2018) that "*there does not seem to exist [*at this day*] a true analogous reverse AD mode to compute generalized derivatives for nonsmooth functions*", illustrating the difficulty of nonsmooth AD. Conservative gradients were introduced as a faithful mathematical model capturing the formal application of calculus rules to subdifferentials by Bolte and Pauwels (2020a;b); Bolte et al. (2021). The author unfamiliar with this notion may reduce, in a ML context, conservative gradients to outputs of calculus rules formally applied to Clarke subgradients and Jacobians. Our goal is to provide an adequate computational complexity theory for conservative calculus, a theory that will therefore match standard practical approaches.

Among other possible first-order options offered by nonsmooth calculus, we also investigate the properties of directional derivatives and those of the Clarke subdifferential. For directional derivatives, our motivation comes from the fact that this nonsmooth operation has general calculus rules, while the Clarke subdifferential is central in terms of variational interpretation.

**Contributions:** The main thesis of this work is that conservative gradients have computational properties similar to smooth derivatives, which are much more favorable than those of alternative nonsmooth oracles such as subgradients or directional derivatives.

• We provide a simple computational model for addressing the question of complexity theory of nonsmooth numerical programs.
• For the backward mode, we prove a *cheap conservative gradient principle* à la Baur-Strassen, generalizing state of the art to nonsmooth programs modeling most NNs. We establish that, regardless of dimension, the computational cost of a conservative gradient is of the order of that of function evaluation. Our results provide a theoretical validation of the fact that the cost of backpropagation does not depend on the programs' smoothness.
• For the forward mode, we relate the computational cost of $p$ directional derivatives to that of $p \times p$ matrix multiplication. We provide lower complexity bounds that illustrate the limits to which this deficiency may be improved. This applies to existing nonsmooth AD frameworks (Khan and Barton, 2012; 2013).
• We establish that computing two distinct elements in the Clarke subdifferential of a given point is NP-hard for simple ReLU programs. This result also applies to the lexicographic subdifferential. In contrast, we show that the problem can be solved in polynomial time for conservative gradients. This reflects the computational difficulty of dealing with the Clarke subdifferential.
• A result of independent interest: deciding differentiability of a ReLU program at a point is NP-hard.

**Relation with existing work:** Conservative gradients were introduced in Bolte and Pauwels (2020a;b) to model "formal subdifferentiation" used by practitioners and nonsmooth "backpropagation". They were further studied in Lewis and Tian (2021); Davis and Drusvyatskiy (2021); Bolte et al. (2021) and empirically investigated in Bertoin et al. (2021). Computational complexity was

only qualitatively considered. We provide a rigorous description of this aspect based an arithmetic computational cost framework capturing programming with nondifferentiable components. The quest for a computationally cheap nonsmooth derivative has a long history in AD literature. Existing works of Griewank (Griewank and Walther, 2008; Griewank, 2013; Griewank and Rojas, 2019; Griewank and Walther, 2020) are essentially based on piecewise smoothness structures (Scholtes, 2012). A cheap subgradient principle was also given in Kakade and Lee (2018), but it requires a very strong qualification condition. As illustrated in Griewank and Rojas (2019), such qualification conditions can be computationally hard to check in practice.

In another research line, based on chain rules for directional derivatives, Khan-Barton (Khan and Barton, 2012; 2013; 2015; Barton et al., 2018) studied the vector forward mode AD. In particular, they investigated the forward AD framework to evaluate elements of the lexicographic subdifferential (see Nesterov (2005)), which is contained in the Clarke subdifferential. In the worst case, the computational overhead ratio they obtain is proportional to the ambient dimension. This contrasts with our cheap gradient principle, whose constant is dimension-less. While these contributions are most relevant to nonsmooth AD, their applicability to large-scale learning models is limited, due to the central role of forward AD.

**Organization of the paper:** We introduce elements of nonsmooth analysis and, in particular, the notion of conservative gradient used throughout this work in Section 2. Section 3 describes a general model of computation that allows one to express the computational cost and complexity of programs, functions and their conservative gradients. This section also presents an abstract program algorithmic differentiation framework. These elements are gathered in Section 4 which presents our extension of the Baur-Strassen result with the cheap conservative gradient principle and its illustrations. To conclude, in Section 5, we describe computational lower bounds for evaluating directional derivatives and distinct subgradients for simple programs.

## 2 NONSMOOTH GENERALIZED GRADIENTS

They are fundamental to expressing variations of nonsmooth losses in Machine Learning. Given a locally Lipschitz continuous function $F : \mathbb{R}^p \to \mathbb{R}$, the *Clarke subdifferential* of $F$ is

$$\partial^c F(x) = \mathrm{conv} \left\{ \lim_{k \to +\infty} \nabla F(x_k) : x_k \in \mathrm{diff}_F, x_k \underset{k \to +\infty}{\to} x \right\} \tag{1}$$

where $\mathrm{diff}_F$ is the full measure set where $F$ is differentiable and $\nabla F$ is the standard gradient (Clarke, 1983). The subdifferential is set-valued, which we write $\partial^c F : \mathbb{R}^p \rightrightarrows \mathbb{R}^p$. For each $x \in \mathbb{R}^p$, elements of $\partial^c F(x)$ are called *Clarke subgradients* of $F$. A selection $d$ in $\partial^c F$, is a function $d : \mathbb{R}^p \to \mathbb{R}^p$ such that for all $x \in \mathbb{R}^p$, $d(x) \in \partial^c F(x)$. If $F$ is $C^1$ then $\partial^c F = \{\nabla F\}$ everywhere so the only possible selection is $d = \nabla F$. We will manipulate derived dictionaries, which typically provide a selection in either the Clarke subdifferential, or more general set-valued maps.

**Example 1** For ReLU: $t \mapsto \max(0, t)$, we have $\partial^c \mathrm{ReLU}(t)$ is $\{0\}$ if $t < 0$, $\{1\}$ if $t > 0$ and $[0, 1]$ if $t = 0$. We may define the function $\mathrm{ReLU}'$ as a selection in $\partial^c \mathrm{ReLU}$ :

$$\mathrm{ReLU}'(t) = 1, \text{ if } t > 0, \qquad \mathrm{ReLU}'(t) = 0, \text{ otherwise.}$$

The chain-rule, essential to AD, generally fails for Clarke subgradients. This is why we now consider the more flexible notion of conservative gradients.

**Definition 1 (Conservative gradient)** Let $F : \mathbb{R}^p \to \mathbb{R}$ be a locally Lipschitz continuous function and $D_F : \mathbb{R}^p \rightrightarrows \mathbb{R}^p$ a locally bounded, nonempty and graph closed set-valued map. Then $D_F$ is a *conservative gradient* for $F$, if for any absolutely continuous curve $\gamma : [0, 1] \to \mathbb{R}^p$,

$$\frac{d}{dt} F(\gamma(t)) = \langle v, \dot{\gamma}(t) \rangle \qquad \forall v \in D_F(\gamma(t)), \qquad \text{for almost all } t \in [0, 1]. \tag{2}$$

In this case, $F$ is called *path differentiable*. Conservative Jacobians are defined similarly. As in Section 2, $d : \mathbb{R}^p \to \mathbb{R}^p$ is a selection of $D_F$ if $d(x) \in D_F(x)$ for all $x \in \mathbb{R}^p$.

A rich class of path differentiable functions is given by locally Lipschitz continuous semi-algebraic functions with the Clarke subdifferential as a conservative gradient. Actually, virtually all functions used in machine learning are path differentiable (Bolte and Pauwels, 2020a;b). The most salient facts about path differentiable functions and their conservative gradients are:

- (**Clarke subgradient**), for all $x \in \mathbb{R}^p$, $\partial^c F(x) \subset \mathrm{conv}(D_F(x))$.
- (**Gradient almost everywhere**) Conservative gradients are gradients a.e (Bolte and Pauwels, 2020a).
- (**First-order oracle**) Selection in conservative gradients can be used as surrogate gradients, while preserving convergence guaranties (Bolte and Pauwels, 2020a;b; Bolte et al., 2021).

Conservative Jacobians can be composed while preserving conservativity (Bolte and Pauwels, 2020a), a feature which do not enjoy Clarke Jacobians: let $F: \mathbb{R}^p \to \mathbb{R}^m$, $G: \mathbb{R}^m \to \mathbb{R}^l$ be locally Lipschitz continuous mappings, $d_F: \mathbb{R}^p \to \mathbb{R}^{m \times p}$ and $d_G: \mathbb{R}^m \to \mathbb{R}^{l \times m}$ be selections in conservative Jacobians for $F$ and $G$ respectively. Then the product mapping $x \mapsto d_G(F(x)) \times d_F(x)$ is a selection in a conservative Jacobian for $G \circ F$. The use of conservative Jacobians provides a very convenient framework to model AD in the nonsmooth case, see Bolte and Pauwels (2020a;b).

A fundamental theorem is the following:

**Theorem 1 (Path differentiable functions are ubiquitous)** (Bolte and Pauwels (2020a)) *Locally Lipchitz semialgebraic (or definable) functions are path differentiable.*

## 3 PROGRAMS, COMPLEXITY AND AUTOMATIC DIFFERENTIATION

### 3.1 CALCULUS MODEL, PROGRAMS, COMPUTATIONAL COST AND COMPLEXITY

A *dictionary* $\mathcal{D}$ is a finite set of real functions (*e.g.* $\{+, -, \times, /\}$), it is paired with $\mathcal{P}^0(\mathcal{D})$, a set of elementary programs implementing them in real arithmetic. Starting from $\mathcal{P}^0(\mathcal{D})$, we aim at capturing the notion of "program of programs" at any depth. As this is an inductive process, we call $k \in \mathbb{N}$ a program "level", which is simply an induction counter needed for consistency. Recursively, programs of level $k + 1$, in $\mathcal{P}^{k+1}(\mathcal{D})$, consist of combinations of outputs of programs of level $k$, in $\mathcal{P}^k(\mathcal{D})$. For example if $P_1$ and $P_2$ are elementary programs in $\mathcal{P}^0(\mathcal{D})$, then the program which sums the outputs of $P_1$ and $P_0$ is of level 1. More precisely:

Let $p, q$ be input and output sizes respectively and $m \geqslant p + q$ a memory size. A *predecessor relation* is a set valued map $\mathrm{pr}: \{1, \ldots, m\} \rightrightarrows \{1, \ldots, m\}$ such that for $i = 1, \ldots, m$
- for $j \in \mathrm{pr}(i)$, $j < i$.
- $\mathrm{pr}(i)$ is empty if $i \leqslant p$ and nonempty otherwise.

An *adapted program sequence* $(g_i)_{i=p+1}^m$ in $\mathcal{P}^k(\mathcal{D})$, is a set of programs such that $g_i$ has $|\mathrm{pr}(i)|$ input arguments and a single output, for all $i = p + 1, \ldots, m$.
Given $(p, q, m, \mathrm{pr}, (g_i)_{i=p+1}^m)$, the program given in Algorithm 1 is a level $k + 1$ *program on* $\mathcal{D}$.

---
**Algorithm 1:**

**Program data:**
$(p, q, m, \mathrm{pr}, (g_i)_{i=p+1}^m)$.

**Input:** $x = (x_1, \ldots x_p)$
1: **for** $i = p + 1, p + 2, \ldots m$ **do**
2:   $x_i = g_i(x_{\mathrm{pr}(i)})$ where
3:   $x_{\mathrm{pr}(i)} = (x_j)_{j \in \mathrm{pr}(i)}$.
4: **end for**
**Return:** $y := (x_j)_{j=m-q+1}^m$.

---

The set of *programs with dictionary* $\mathcal{D}$ is $\mathcal{P}(\mathcal{D}) = \bigcup_{k \geqslant 0} \mathcal{P}^k(\mathcal{D})$. We shall see however that $\mathcal{P}^k(\mathcal{D}) = \mathcal{P}^1(\mathcal{D})$ for all $k$, using modification of the computational graph.

A *cost on a dictionary* $\mathcal{D}$ is a nonnegative function on $\mathcal{D}$, it extends additively by induction on programs on $\mathcal{D}$ through the rule $\mathrm{cost}(P) = \sum_{i=1}^m \mathrm{cost}(g_i)$ where $P$ is a program on $\mathcal{D}$ as described in Algorithm 1. A direct example is the dictionary of arithmetic functions $\{+, -, \times, /\}$, together with addition or multiplication by fixed constants, denoted by $+c$ and $\times c$ respectively[1], see also Section A.1. Throughout the paper, we assume that dictionaries contain at least operations $+$ and $\times$.

Each program on $\mathcal{D}$ may be represented by a program in $\mathcal{P}^1(\mathcal{D})$ with the same cost, by expanding all subprograms until they reduce to an elementary program. Cost evaluation is thus well defined on such programs. As detailed in Appendix A.1, this model of computation is equivalently expressed using directed acyclic graphs.

---
[1] Constants need to be distinguished from variables (for instance to define a polynomial)

To sum up, we have defined the set of programs $\mathcal{P}(\mathcal{D})$ on $\mathcal{D}$, which includes programs of programs. The programs $g_i$ in Algorithm 1 may be taken in $\mathcal{P}(\mathcal{D})$. The cost of a program is evaluated through the calls it makes to elementary programs in the dictionary.

**Programs vs functions:** A program $P$ defines a unique input-output function $f$: we say that $P$ "computes" $f$, or "implements" $f$, and with a slight abuse of notation, we will identify $P$ and $f$ when there is no ambiguity (*e.g.* derivative of $P$). We use the equivalence relation $\sim$ to relate programs computing the same function. The equivalence classes correspond to functions expressible by programs with a given dictionary $\mathcal{D}$. Given a function $f\colon \mathbb{R}^p \to \mathbb{R}^q$ and a program $P$ on dictionary $\mathcal{D}$, with $p$ inputs and $q$ outputs, we write $f = [P]$ to denote the fact that $P$ is in the equivalence class of programs computing $f$, that is, $P$ implements $f$.

**Complexity of a function:** The complexity of a function $f$ over a dictionary $\mathcal{D}$ is the quantity $\mathrm{comp}(f, \mathcal{D}) = \inf \{\mathrm{cost}(P), \ s.t \quad P \in \mathcal{P}(\mathcal{D}), \ f = [P]\}$, the infimum being over all programs implementing $f$ on dictionary $\mathcal{D}$. It could be infinite, if it is finite then it is attained.

## 3.2 Automatic differentiation

We pertain to programs implementing functions, that is Algorithm 1 with single outputs $q = 1$.

Given a dictionary $\mathcal{D}$ of locally Lipschitz path differentiable functions, a *derived dictionary* is a set of functions $\mathcal{D}' \supset \mathcal{D}$ which extends $\mathcal{D}$ and contains operations required to express at least an element in a conservative gradient for each of the functions in $\mathcal{D}$, for example, an element in the Clarke subdifferential. We also consider a cost function on $\mathcal{D}'$, which we denote by cost and which extends to programs over $\mathcal{D}'$. Given programs $g_i$ on $\mathcal{D}$, $i = p + 1, \ldots, m$, we define $d_i$ a *derived program* on $\mathcal{D}'$, with $|\mathrm{pr}(i)|$ inputs and outputs, which returns an element of a conservative gradient for $g_i$ (as for instance a Clarke subgradient, or simply a gradient in the $C^1$ case). By $gd_i$, we denote a program on $\mathcal{D}'$ evaluating $(g_i(x), d_i(x))$ jointly for a given $x$. We denote by Algorithm 1', an extension of Algorithm 1 which additionally returns $w_i = d_i(x_{\mathrm{pr}(i)})$ for $i = p + 1, \ldots, m$, by replacing line 2 in Algorithm 1 with a call to $gd_i$ instead of $g_i$. The backward (resp. forward) AD program $\mathrm{backprop}(P)$ (resp. $\mathrm{forprop}(P)$) is defined as follows:

---

**Algorithm 2:** Algorithmic differentiation of $P$ as in Section 3.1

**Input:** variables $(x_i)_{i=1}^p$
**Forward evaluation with derivatives:** evaluate $w_i = d_i(x_{\mathrm{pr}(i)})$, $i = p + 1, \ldots, m$,
with Algorithm 1': Algorithm 1 with $gd_i$ instead of $g_i$ on line 2.

1: **Forward mode:**
2: Initialize: $\frac{\partial x_i}{\partial x} = e_i$ , $i = 1, \ldots, p$, from canonical basis in $\mathbb{R}^p$.
3: **for** $i = p + 1, \ldots m$ **do**
4:
$$\frac{\partial x_i}{\partial x} = \sum_{j \in \mathrm{pr}(i)} \frac{\partial x_j}{\partial x} w_i[j]$$
where $x = (x_1, \ldots, x_p)$.
5: **end for**
**Return:** $\frac{\partial x_m}{\partial x}$ and eventually $x_m$.

1: **Backward mode:**
2: Initialize: $v = e_m$
3: **for** $t = m, \ldots p + 1$ **do**
4:     **for** $j \in \mathrm{pr}(t)$ **do**
5:         Update coordinate $j$ of $v$:
$$v[j] := v[j] + v[t]w_t[j]$$
6:     **end for**
7: **end for**
**Return:** $(v[j])_{j=1}^p$ and eventually $x_m$.

---

Note that Algorithm 2 starts with Algorithm 1', i.e., Algorithm 1 with $gd_i$ instead of $g_i$ on line 2. Its computational cost, denoted $\mathrm{cost}(gd_i)$, should be thought of as an exogenous parameter: *it may model, for instance, the use of underlying software libraries or the hardware properties.*

## 4 Computational complexity of Nonsmooth AD

We now evaluate the complexity of the $\mathrm{forprop}$ and $\mathrm{backprop}$ operations for conservative gradients in the path-differentiable case – which encompasses, as mentioned earlier, all semi-algebraic and

definable locally Lipschitz functions. We show, in particular, that backpropagation with conservative gradients has a computational overhead ratio that is independent of the dimension. This is in contrast with the best known algorithmic oracles for the Clarke subdifferential (see Khan and Barton (2012; 2013; 2015); Barton et al. (2018) and Appendix A.2), whose computational overhead ratio scales linearly with the dimension.

**Theorem 2 (Complexity of nonsmooth AD)** *Let $P$ be a program over a dictionary $\mathcal{D}$ of path-differentiable functions with $p$ inputs as in Algorithm 1 & 2. Then, the corresponding function $[P]$ is path differentiable, there is a conservative gradient $D_P$ for the function $[P]$ such that:*

*(i) (**Cost of backward mode**) At each input point $x \in \mathbb{R}^p$, the output of program $\mathrm{backprop}(P)$ is in $D_P(x)$ and we have $\mathrm{cost}(\mathrm{backprop}(P)) \leqslant \omega_b \, \mathrm{cost}(P)$, where*

$$\omega_b = \max_{i=p+1,m} \left\{ (\mathrm{cost}(gd_i) + 2\max(\mathrm{cost}(+), \mathrm{cost}(\times))|\mathrm{pr(i)}|) \, / \, \mathrm{cost}\,(g_i) \right\}. \tag{3}$$

*(ii) (**Cost of forward mode**) At each input point $x \in \mathbb{R}^p$, the output of program $\mathrm{forprop}(P)$ is in $D_P(x)$ and we have $\mathrm{cost}(\mathrm{forprop}(P)) \leqslant \omega_f \times \mathrm{cost}(P)$ where*

$$\omega_f = \max_{i=p+1,m} \left\{ (\mathrm{cost}\,(gd_i) + p|\mathrm{pr(i)}|\mathrm{cost}(\times) + p(|\mathrm{pr(i)}| - 1)\mathrm{cost}(+)) \, / \, \mathrm{cost}\,(g_i) \right\}.$$

There is a dissymmetry between the two modes since the constant $\omega_b$ is independent of the dimension $p$. This is why property (i) is sometimes called the "cheap conservative gradient principle" extending the classical smooth one which was derived by Baur and Strassen (1983) for real rational functions. Theorem 2 describes worst case upper bounds (maximum over $i$), which are tight, for example if $pr(i)$, costs of $g_i$ and $gd_i$ are independent of $i$.

We will consider several examples now.

**The class of ReLU programs** Let $\mathcal{D}_{\mathrm{ReLU}}$ be the dictionary composed of elementary arithmetic operations, logarithm, exponential and the ReLU function:

$$\mathcal{D}_{\mathrm{ReLU}} := \{+, \times, +c, \times c, \mathrm{inv}, \exp, \log, \mathrm{ReLU}\}. \tag{4}$$

A ReLU program $P$ is a program with dictionary $\mathcal{D}_{\mathrm{ReLU}}$; it can be expressed in a compositional form (Section 3.1) with program sequences in $\mathcal{D}_{\mathrm{ReLU}}$. Note that this yields path differentiable functions.

**Assumption 1 (Computational Cost)** In Algorithms (2), define the dictionary $\mathcal{D}'_{\mathrm{ReLU}} := \mathcal{D}_{\mathrm{ReLU}} \cup \{\mathrm{ReLU}'\}$ as in Example 1; then, all operations from $\mathcal{D}'_{\mathrm{ReLU}}$ have unit cost (see Remark 1).

**Corollary 1 (Backprop complexity of ReLU programs)** *Let $P$ be a ReLU program, under Assumption 1, we have: $\mathrm{cost}(\mathrm{backprop}(P)) \leqslant 5\mathrm{cost}(P)$. This extends to more complex cost weighting schemes (Remark 1) and to selection functions which virtually capture all losses in ML (Remark 2).*

Table 1: Complexity constant of $\omega_b$ in Theorem 2 for elementary $g$ in $\mathcal{D}_{\mathrm{ReLU}}$ and derived program with dictionary $\mathcal{D}'_{\mathrm{ReLU}}$. This proves Corollary 1 (more details in appendix B.1).

| $g$ | $(+, \times)$ | $(+c, \times c)$ | $\log$ | $\exp$ | $\mathrm{inv}$ | $\mathrm{ReLU}$ |
|---|---|---|---|---|---|---|
| $(\mathrm{cost}(gd) + 2\mathrm{cost}(\times)|\mathrm{pr}|) \, / \, \mathrm{cost}\,(g)$ | 5 | 3 | 4 | 3 | 5 | 3 |

**Remark 1 (On refined cost systems)** Unit cost in Assumption 1 gives a simple interpretation to Corollary 1: the cost of a program is the total number of numerical operations. This rough estimate of computational complexity, could be refined with different weighting schemes. However, the obtained constant 5 is robust to many different weighting choices, far beyond Assumption 1. We detail an example in the Appendix B.2 for which the cost of all smooth nonlinear operations different from $+$ or $\times$ is $c_{\mathrm{nonlin}} \geqslant 1$ and we model the cost of sign branching in computation of ReLU and ReLU$'$ with constant $c_{\mathrm{ReLU}} \geqslant 0$. This yields the same constant as in Corollary 1.

**Remark 2 (Beyond ReLU programs)** Many other dictionaries could be considered. ReLU is an example chosen for its simplicity, but Corollary 1 would hold similarly (with the same constant 5) for many different nonsmooth activations or components such as absolute value, max-pooling, ELU function, $\ell_1$ and $\ell_\infty$ norms. Similar results could be developed for the class of selection functions, which encompasses the vast majority of ML building blocks (see Bolte and Pauwels (2020b)). This is sketched in Appendix B.3.

**Chaining backpropagation derived programs**  Our approach is flexible enough to describe "programs of programs" and backpropagation chaining. Let $P$ be a program as in Algorithm 1, with adapted ReLU program sequence $\{(g_i)_{i=p+1}^m\}$. If $\text{cost}(g_i) \gg |\text{pr}(i)|$, $g_i$ is a "long program", with many operations per input. We may set $gd_i = \text{backprop}(g_i)$ using Algorithm 2, $i = p+1, \ldots, m$. From Corollary 1, we have $\text{cost}(gd_i)/\text{cost}(g_i) \leqslant 5$, and for long programs $\omega_b \simeq 5$ in Theorem 2. This illustrates the versatility of our approach as it captures the complexity of chaining $\text{backprop}$ operations, the resulting estimate being quite sharp in the regime of long programs.

**Beyond backpropagation**  Programs may be differentiated by other means than backpropagation. Examples include, forward propagation, with applications in optimization and algorithmic unrolling (Mehmood and Ochs, 2020; Lorraine et al., 2020; Maclaurin et al., 2015), implicit differentiation Agrawal et al. (2018); Winston and Kolter (2020); Bai et al. (2019); Bolte et al. (2021) with application in optimization and hyperparameter optimization (Bertrand et al., 2020), adjoint differentiation (Plessix, 2006) in programs with components involving ordinary differential equations (Courtier and Rabier, 1997; Chen et al., 2018), differentiation of conjugate gradient (Gratton et al., 2014), Cholesky algorithm (Smith, 1995), approximation of Jacobian matrices involving a non-uniform FFT (Wang and Fessler, 2021).

Let $P$ be a program as in Algorithm 1. Theorem 2 relates the complexity of combining derived programs in Algorithm 2 to the following quantities, for $i = p+1, \ldots, m$:

- $\text{cost}(gd_i)/\text{cost}(g_i)$: the "computational overhead ratio".
- $|\text{pr}(i)|\text{cost}(\times)/\text{cost}(g_i)$: the ratio between multiplication cost and average cost per input argument.

The first quantity depends on the technique used to obtain $gd_i$. The second quantity is typically less than 2 (at least one arithmetic operation per input) and becomes negligible for long programs (many operations per input).

For example in Mehmood and Ochs (2020); Lorraine et al. (2020); Maclaurin et al. (2015), for one $i$, the program $gd_i$ is an optimization algorithm in $\mathbb{R}^p$, a long program differentiated using forward propagation. The corresponding overhead ratio is in this case $3p + 5$ (Theorem 2). If combined with an outer backward pass, we obtain a dimension-dependent overhead ratio, in contrast with full backward differentiation. Our model provides computational cost estimates for mixed techniques, here a combination of inner forward and outer backward propagation.

## 5 ON THE COMPUTATIONAL HARDNESS OF GENERALIZED GRADIENTS

Let $P$ and $DP$ be two programs such that $DP$ evaluates jointly $P$ and a derivative of $P$. In the sequel, we use the term (*computational*) *overhead ratio* of $DP$ to denote the quantity $\frac{\text{cost}(DP)}{\text{cost}(P)}$ and computational overhead ratio of derivatives of $P$ to denote the quantity $\frac{\text{comp}(DP)}{\text{cost}(P)}$. As established in Theorem 2, this ratio is dimensionless in the case of backpropagation with conservative gradients. Are there other ways to compute cheap nonsmooth gradients? Toward an answer to this question, we discuss this ratio for other nonsmooth differentiation oracles: directional derivatives (for which we relate worst-case complexity to that of matrix multiplication), lexicographic derivatives with forward AD (with an overhead ratio of order $p$ Barton et al. (2018)). As for the Clarke subdifferential, we prove the hardness of subgradients enumeration. Our motivation to estimate the complexity of these particular types of derivatives (directional, lexicographic and Clarke) is that they serve as a basis to alternative implementable AD approaches (see Barton et al. (2018) and references therein), and are thus concurrent strategies of conservative gradient backpropagation. The results presented below do not provide a definitive answer, but they strongly suggest that backpropagation of conservative gradients has a much more favorable complexity.

### 5.1 THE OVERHEAD RATIO FOR EVALUATING $p$ DIRECTIONAL DERIVATIVES

Given $G \colon \mathbb{R}^p \to \mathbb{R}$ locally Lipschitz and $x, d \in \mathbb{R}^p$, the directional derivative of $G$ at $x$ in direction $d$ is given by $\lim_{t \downarrow 0}(G(x + td) - G(x))/t$ when the limit exists. This section considers a family of functions with $p$ inputs and $q$ real parameters, represented by a locally Lipschitz function $F \colon \mathbb{R}^p \times \mathbb{R}^q \to \mathbb{R}$, for which we investigate hardness of evaluation of $p$ directional derivatives. The function $F$ may describe, for instance, a ReLU feedforward neural network empirical loss, parameterized by

$q$ real weights, with $p$ inputs. For functions represented by $\mathrm{ReLU}$ programs, we prove an overhead ratio of order $p^{\omega-2+o(1)}$ where $\omega$ is the matrix multiplication exponent (see definition below). In all rigor, it is not known whether $\omega > 2$ or $\omega = 2$, so the derived ratio could be essentially dimensionless (if $\omega = 2$), though all practical evidences are against this so far. The best known lower bound is $\omega < 2.37$, and in practice, the matrix multiplication exponent is closer to 2.7, both corresponding to a dimension-dependent overhead, in contrast with the smooth case with essentially dimensionless overhead ratio to evaluate $p$ directional derivatives (essentially a gradient).

**Complexity of matrix multiplication:** Throughout this section, we set $\mathcal{D} = \{+, \times, +c, \times c\}$, with unit costs (corresponding to polynomial functions). Denote by $c(p)$ complexity of $p \times p$ matrix multiplication. More precisely, if $f \colon \mathbb{R}^{p \times p} \times \mathbb{R}^{p \times p} \to \mathbb{R}^{p \times p}$ is such that $f(A, B) = AB$ for all, square matrices $A, B \in \mathbb{R}^{p \times p}$, we have $c(p) = \mathrm{comp}(f, \mathcal{D})$, which we may write $c(p) = p^{\omega+o(1)}$ where $\omega$ is called the *matrix multiplication exponent*. Note that $c(p) \geqslant p^2$, as one needs at least one operation for each of the $2p^2$ entries.

**Directional derivatives:** Given a function $F \colon \mathbb{R}^p \times \mathbb{R}^q \to \mathbb{R}$, we denote by $F_1' \colon \mathbb{R}^p \times \mathbb{R}^q \times \mathbb{R}^{p \times p} \to \mathbb{R}^p$ the function which associates to $x \in \mathbb{R}^p$, $y \in \mathbb{R}^q$ and a matrix $A \in \mathbb{R}^{p \times p}$ the $p$ directional derivatives with respect to $x$ variable, for fixed $y$, in directions given by the columns of $A$. The proof of the following theorem is given in Section C.

**Theorem 3 (Computational ratio for directional derivatives)** *There exists a function $F \colon \mathbb{R}^p \times \mathbb{R}^q \to \mathbb{R}$ and a program $P_F$ implementing $F$ on dictionary $\{+, \times, \mathrm{ReLU}, +c, \times c\}$ (all operations have unit cost), such that for any program $P'$ implementing $(y, A) \mapsto F_1'(0, y, A)$ on derived dictionary $\{+, \times, \mathrm{ReLU}, \mathrm{ReLU}', +c, \times c\}$,*

$$\mathrm{cost}(P')/\mathrm{cost}(P_F) \geqslant (c(p) - 5p)/(40p^2) = p^{\omega-2+o(1)}. \tag{5}$$

Theorem 3 has $q$ parameters, parametric dependency is required to express hardness. Indeed, for some parameter values, computation may be trivial (e.g. null values). Alternatively, it states that for some values of the $q$ parameters, computing $p$ directional derivatives has cost as in (5).

The bound in (5) is sharp up to multiplicative constants for linear ReLU networks, see Remark 5 in Appendix A.2.

**Consequences:** Our overhead estimate is roughly $p^{\omega-2}$, it constitutes a bottleneck: a "cheap nonsmooth $p$ directional derivatives principle", would imply easy matrix multiplication, to the point that $\omega = 2$. Since the seminal work of Strassen et al. (1969), it is known that $\omega \leqslant \log_2(7) \simeq 2.81$. Determining the precise exponent $\omega$ has been an object of intense research Robinson (2005). Asymptotically, one has $2 \leqslant \omega < 2.373$, see Williams (2012); Le Gall (2014), the best known bound being given in Alman and Williams (2021). In this case, the estimate in (5) is roughly $p^{0.373}$.

These estimates may involve non-constructive existence proofs, or suffer from the curse of recursion: meaningful efficiency occurs only for inaccessible sizes. According to Dumas and Pan (2016), for values $p \leqslant 10^6$ the most efficient practical algorithms have a complexity of the order $p^{2.774}$, resulting in an overhead of order $p^{0.774}$, in contrast with the constant overhead incurred by nonsmooth backpropagation. More discussion is given in Appendix A.2.

**Comparison with the smooth case:** If $F$ is $C^1$, evaluating $p$ directional derivatives is comparatively easier because $F'(x, d) = \langle \nabla F(x), d \rangle$ for all $x, d \in \mathbb{R}^p$. Hence, one may first evaluate $\nabla F$ (once), at a cost similar to that of $F$ (cheap gradient principle), and then evaluate $p$ scalar products, at a cost $p^2$. If the cost of $F$ is of order $p^2$ at least (for example $F$ is a feedforward neural network with $p$ inputs and a layer of $p$ hidden neurons), then this is overall proportional to the cost of computing $F$.

### 5.2 Computing Clarke subgradients using forward automatic differentiation

In Khan and Barton (2012; 2013; 2015), several automatic differentiation strategies are proposed to evaluate elements of the Clarke subdifferential. These approaches are based on directional (Shapiro, 1990) and lexicographic derivatives (Nesterov, 2005) which satisfy a chain rule under structural assumptions. The chain rule may be implemented using the vector forward mode of automatic differentiation (Barton et al., 2018), which suffers from computational overhead scaling linearly

in $p$, contrary to the reverse mode in Theorem 2. Reducing this factor is an open question, even for compositional functions involving only univariate nonsmoothness such as absolute value (Khan, 2018). More details are given in Appendix A.2.1.

## 5.3 COMPUTATIONAL HARDNESS OF SUBGRADIENT ENUMERATION

We investigate in this section the hardness finding subgradients for programs defined on the elementary dictionary $\mathcal{D}_0 = \{+, -, \text{ReLU}\}$ with unit costs. Let us denote by $\mathcal{P}(\mathcal{D}_0)$ the set of such programs. We will, with a slight abuse of notation, identify a program $P \in \mathcal{D}_0 = \{+, -, \text{ReLU}\}$ with the function it computes to state our complexity result (proof in Section D).

**Theorem 4 (Clarke subgradients and NP-Hardness)**
*(i) The problem of finding two distinct subgradients in the Clarke subdifferential of $P \in \mathcal{P}(\mathcal{D}_0)$ at given input (or one single subgradient if it is reduced to a singleton) is NP-hard.*
*(ii) Deciding if $P \in \mathcal{P}(\mathcal{D}_0)$ is not differentiable at some given input is NP-hard.*

**Remark 3** In Theorem 4, numerical parameters and inputs are constrained to be in $\{-1, 0, 1\}$, so that the hardness result does not depend on numerical representation and only involves program size (strong NP-hardness). See Appendix D for more details.

The above problems (i)-(ii) enter the field of computational complexity as we consider programs $P \in \mathcal{P}(\mathcal{D}_0)$ with a natural notion of size, given by their cost, $\text{cost}(P)$, the number of operations (recall that we assumed unit costs). Since the considered programs implement piecewise linear functions, it follows from (Barton et al., 2018, Proposition 2.7) that, our hardness result also holds for the lexicographic subdifferential Nesterov (2005), which reduces in this case to the set of neighboring gradients (see Section D).

The counterpart of the above problem for AD conservative gradients as in Definition 2 is tractable, illustrating a major computational difference between Clarke subdifferential and AD conservative gradient. The proof is in Section D.4, by reduction to a graph shortest path problem.

**Proposition 1 (Finding two elements in autodiff conservative gradients is tractable)** *Given $P \in \mathcal{P}(\mathcal{D}_0)$, with conservative gradient $D_P$ given by Theorem 2, finding two elements in $D_P(x)$ at a given input $x$ (or one single element if $D_P(x)$ is a singleton) is solvable in polynomial time.*

## 6 CONCLUSION

We extended the "cheap gradient" principle to nonsmooth automatic differentiation with a flexible version of Baur-Strassen's result: the overhead ratio of conservative gradients is independent of the dimension. On the other hand, we showed that the potential gain in efficiency of forward AD for multiple directional derivatives is limited due to an intrinsic connection to matrix multiplication. Finally, we have shown that for simple ReLU networks, the enumeration of Clarke subgradients is computationally hard, in contrast to the enumeration of conservative gradients.

The global picture is significantly different from the smooth case, with a well understood "cheap gradient" principle that yields "cheap $p$ directional derivatives", illustrating the specificities of nonsmoothness. Our results confirm the centrality of conservative gradients in nonsmooth AD and machine learning: they generalize gradients with a clear "cheap principle", contrary to concurrent notions. An important open question in this context is the complexity of subgradients, or, in other words, the existence of a "cheap subgradient principle". We conjecture a negative answer in general.

## ACKNOWLEDGMENTS AND DISCLOSURE OF FUNDING

The authors acknowledge the support of the AI Interdisciplinary Institute ANITI funding under the grant agreement ANR-19-PI3A-0004. The authors acknowledge the support of the Association nationale de la recherche et de la technologie (ANRT) and Thales LAS France, which contributed to Ryan B's grant. Jérome B. and Edouard P. acknowledge the financial support of Air Force Office of Scientific Research, Air Force Material Command, USAF, under grant numbers FA9550-19-1-7026 FA8655-22-1-7012, and ANR MaSDOL 19-CE23-0017-01. Jérôme B. also acknowledges the

support of ANR Chess, grant ANR-17-EURE-0010, TSE-P and the Centre Lagrange. We thank our collaborators in the Thales LAS France, especially Andrei Purica, for helpful comments. We are grateful to Serge Gratton, Pierre Weiss and Pierre Boudier for useful reference suggestions.

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

This is the appendix for "On the complexity of nonsmooth automatic differentiation".

## CONTENTS

## A   FURTHER COMMENTS, DISCUSSION AND TECHNICAL ELEMENTS

### A.1   COMMENTS ON SECTION 3

#### A.1.1   COMPUTATIONAL MODEL IN SECTION 3.1

**DAG representation and examples 3.1:** We start with a remark regarding representations of programs as directed acyclic graphs and use them to illustrate the model of computation proposed in the main text. It reduces to that of arithmetic circuit complexity for a dictionary composed of elementary arithmetic operations.

**Remark 4 (Programs as directed graphs)** A predecessor relation trivially describes a directed acyclic graph (DAG). Therefore, a program is equivalently represented as a DAG, nodes corresponding either to input variables (empty predecessor) or computation (nonempty predecessor). Directed edges connect predecessor nodes to their successors. Each computation node contains a lower-level program (with a single output), with the number of input edges being coherent with the number of arguments. The cost of a node is that of the underlying program and the cost of $P$ is the sum of the costs of its nodes. Nodes without outer edges are output nodes. See examples in Appendix A.1.

We represent programs using the DAG representation as in Remark 4. Let us define a simple dictionary $\mathcal{D} := \{+, \times\}$ and introduce a level 0 elementary program $P_0$ such that $P_0(a, b) = a + b$ meaning that $P_0$ computes the quantity $a + b$. $P_0$ is identified with $+$ from the dictionary. We also introduce a level 1 program $P_1$ such that $P_1(a, b, c) = a \times (b + c)$. We can construct an equivalent level 1 program, $Q_1$ such that $Q_1(a, b, c) = a \times b + a \times c$, in this case, we have $P_1 \sim Q_1$, or $[P_1] = [Q_1]$ since they compute the same quantity. The level 2 program $P_2$ is such that $P_2(a, b, c, d) = (a + b) \times (c + d) = Q_1(a, c, d) + P_1(b, c, d)$ and uses level 1 programs $Q_1$ and $P_1$ in its computation nodes. The Directed Acyclic Graphs (DAGs) representing these programs are given in Figure 1. Assuming $\mathrm{cost}(+) = \mathrm{cost}(\times) = 1$, we have $\mathrm{cost}(P_0) = 1$, $\mathrm{cost}(P_1) = 2$, $\mathrm{cost}(Q_1) = 3$ and $\mathrm{cost}(P_2) = \mathrm{cost}(Q_1) + \mathrm{cost}(P_1) + \mathrm{cost}(\times) = 6$.

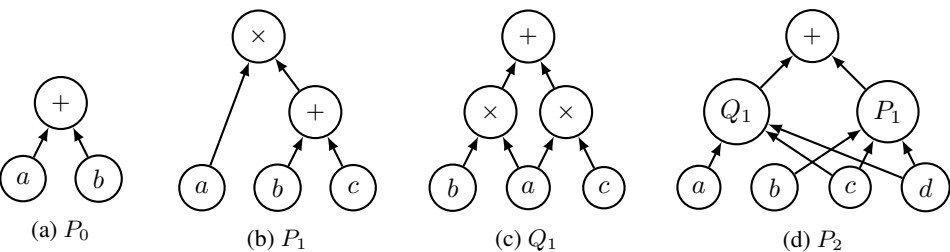

    (a) $P_0$         (b) $P_1$         (c) $Q_1$         (d) $P_2$

Figure 1: DAG illustrating different programs with dictionary $\mathcal{D} := \{+, \times\}$. (a) $P_0(a, b) = a + b$, of level 0 which is identified with $+$ from the dictionary, (b) $P_1(a, b, c) = a(b + c)$, of level 1, (c) $Q_1(a, b, c) = ab + ac$, of level 1 and equivalent to $P_1$, (d) $P_2(a, b, c, d) = (a + b)(c + d) = Q_1(a, c, d) + P_1(b, c, d)$, of level 2.

## A.2 COMMENTS ON SECTION 5

### A.2.1 FORWARD AD AND CLARKE SUBGRADIENTS

Nesterov (2005) introduced the notion of lexicographic subdifferential, denoted here $\partial_L F$ for a Lipschitz function $F\colon \mathbb{R}^p \to \mathbb{R}$. The construction of $\partial_L F$ is based on successive local approximations of $F$ with directional derivatives, and one has $\partial_L F(x) \subset \partial^c F(x)$ for all $x$ such that the first term is well defined.

It is known that automatic differentiation can be used to compute directional derivatives, particularly the forward mode of automatic differentiation (Griewank and Walther, 2008). Based on this observation, Khan and Barton developed several algorithms to evaluate elements of $\partial^c F$, based on directional derivatives (Khan and Barton, 2012; 2013; 2015). They concentrate on piecewise $C^1$ functions, see for example Scholtes (2012), and propose to handle compositional structures with different restrictions on the function class considered, such as functions in abs-normal forms (Khan and Barton, 2012), or broader classes (Khan and Barton, 2013; Barton et al., 2018).

All these procedures either require to evaluate $p$ directional derivatives (Khan and Barton, 2012; 2013), or rely on forward chain rule propagation for lexicographic derivatives (Khan and Barton, 2015; Barton et al., 2018), which also require to maintain $p$ directional derivatives. For this reason, all these methods suffer from a multiplicative computational overhead ratio of the order of $p$ in the worst case, and it is not known if this could be improved (Barton et al., 2018), although efforts have been made in this direction (Khan, 2018).

### A.2.2 MATRIX MULTIPLICATIONS

**Remark 5** The lower bound described in Theorem 3 is sharp for a linear ReLU network $F$ as in (11) involving only square $p \times p$ matrices. Indeed, $p$ directional derivatives of $F$ in directions $a_1, \ldots, a_p$, can be computed with roughly $Lc(p)$ operations, using a matrix multiplication algorithm realizing the $c(p)$ bound, for example using the forward mode of AD Khan and Barton (2012; 2013). The naive $P_F$ algorithm for forward evaluation performs roughly $2Lp^2$ operations which results in the bound (neglecting terms of order one in numerator and denominator),

$$\frac{\text{comp}(F_d, \mathcal{D} \cup \{\text{ReLU}, \text{ReLU}'\})}{\text{cost}(P_F)} \leqslant \frac{c(p)}{2p^2},$$

for this class of networks, to be compared with (5). Finally, we remark that in the smooth case such complexity estimates reduce to gradient computation which can be done using backward algorithmic differentiation with a constant multiplicative overhead ratio.

We denote by $F_d$, the function $F_d\colon (y, A) \mapsto F_1'(0, y, A)$ which computes $p$ directional derivatives at a given point. Setting $\omega = \limsup_{p\to\infty} \log(c(p))/\log(p)$, since $P'$ is an arbitrary program implementing $F_d$, we have shown that asymptotically, for any $\epsilon > 0$

$$\sup_{p, F=[P_F], P_F \in \mathcal{P}(\mathcal{D} \cup \{\text{ReLU}\})} \frac{\text{comp}(F_d, \mathcal{D} \cup \{\text{ReLU}, \text{ReLU}'\})}{\text{cost}(P_F)} \times p^{2-\omega+\epsilon} = +\infty,$$

where the supremum is taken over all $p$ and all functions $F\colon \mathbb{R}^{p \times q} \to \mathbb{R}$ implemented by a program $P_F$ with dictionary $\mathcal{D} \cup \{\text{ReLU}\}$. It is not known whether $\omega > 2$.

## B PROOFS RELATED TO SECTION 4

**Proof of Theorem 2:** Given a program $P$ as in Section 3.1, the path differentiability of $[\mathcal{P}]$ is immediate by composition and the chain rule property. The associated conservative gradient $D_P$ is constructed in Bolte and Pauwels (2020a).

We have the following cost estimates which can be deduced from the definition of the cost of a program in Section 3.1.

- **Algorithm 1** forward evaluation:

$$\text{cost}(P) = \text{cost}(\text{Algorithm 1}) = \sum_{i=p+1}^{m} \text{cost}(g_i) \tag{6}$$

- **Algorithm 1** forward evaluation with derivatives: Algorithm 1' with $gd_i$ instead of $g_i$ on line 2

$$\text{cost}(\text{Algorithm 1'}) = \sum_{i=p+1}^{m} \text{cost}\,(gd_i) \tag{7}$$

- **Algorithm 2** backward AD cost:

$$
\begin{aligned}
\text{cost}(\text{backprop}(P)) =\ & \text{cost}(\text{Algorithm 1'}) + \sum_{i=p+1}^{m} |\texttt{pr}(i)|(\text{cost}(+) + \text{cost}(\times)) \\
=\ & \sum_{i=p+1}^{m} \text{cost}\,(gd_i) + |\texttt{pr}(i)|(\text{cost}(+) + \text{cost}(\times)).
\end{aligned}
\tag{8}
$$

- **Algorithm 2** forward AD cost:

$$
\begin{aligned}
\text{cost}(\text{forprop}(P)) =\ & \text{cost}(\text{Algorithm 1'}) + \sum_{i=p+1}^{m} p|\texttt{pr}(i)|\text{cost}(\times) + p(|\texttt{pr}(i)| - 1)\text{cost}(+) \\
=\ & \sum_{i=p+1}^{m} \text{cost}\,(gd_i) + p|\texttt{pr}(i)|\text{cost}(\times) + p(|\texttt{pr}(i)| - 1)\text{cost}(+).
\end{aligned}
\tag{9}
$$

Let us derive the complexity bound of Algorithm 1 according to Algorithm 2.

**Backward AD complexity result:**  Using (8) and the fact that $\text{cost}$ has value in $\mathbb{R}_+^*$, we have

$$
\begin{aligned}
\text{cost}(\text{backprop}(P)) &= \sum_{i=p+1}^{m} \text{cost}\,(gd_i) + |\texttt{pr}(i)|(\text{cost}(+) + \text{cost}(\times)) \\
&= \sum_{i=p+1}^{m} \text{cost}(g_i) \times \frac{\text{cost}\,(gd_i) + |\texttt{pr}(i)|(\text{cost}(+) + \text{cost}(\times))}{\text{cost}(g_i)} \\
&\leqslant \max_{i=p+1,m} \left( \frac{\text{cost}\,(gd_i) + |\texttt{pr}(i)|(\text{cost}(+) + \text{cost}(\times))}{\text{cost}(g_i)} \right) \sum_{i=p+1}^{m} \text{cost}(g_i),
\end{aligned}
$$

where the inequality is due to factorization by the maximal value. Using (6), we obtain

$$\text{cost}(\text{backprop}(P)) \leqslant \omega_b \times \text{cost}(P)$$

where $\omega_b$ is given in (3). This proves point (i).

**Forward AD complexity result:**  Using (9) and the fact that $\text{cost}$ has value in $\mathbb{R}_+^*$, we have

$$
\begin{aligned}
\text{cost}(\text{forprop}(P)) &= \sum_{i=p+1}^{m} \text{cost}\,(gd_i) + p|\texttt{pr}(i)|\text{cost}(\times) + p(|\texttt{pr}(i)| - 1)\text{cost}(+) \\
&= \sum_{i=p+1}^{m} \text{cost}(g_i) \times \frac{\text{cost}\,(gd_i) + p|\texttt{pr}(i)|\text{cost}(\times) + p(|\texttt{pr}(i)| - 1)\text{cost}(+)}{\text{cost}(g_i)} \\
&\leqslant \max_{i=p+1,m} \left( \frac{\text{cost}\,(gd_i) + p|\texttt{pr}(i)|\text{cost}(\times) + p(|\texttt{pr}(i)| - 1)\text{cost}(+)}{\text{cost}(g_i)} \right) \times \\
&\qquad \sum_{i=p+1}^{m} \text{cost}(g_i),
\end{aligned}
$$

where the inequality is due to factorization by the maximal value. Using (6), we obtain

$$\text{cost}(\text{forprop}(P)) \leqslant \omega_f \times \text{cost}(P)$$

where $\omega_f$ is given in (3).

$\square$

### B.1 Justification of the complexity Table 1 of the $\mathcal{D}_{\mathrm{ReLU}}$-Dictionary.

The proof of Corollary 1 follows from Theorem 2 by computing the relevant constants. They are shown in Table 1, let us justify the proposed numbers.

**Case 1** ($\mathrm{cost}(\times), \mathrm{cost}(+)$) *Let us define $g(a, b) = a \times b$. To evaluate $g$, we need one operation from $\mathcal{D}_{\mathrm{ReLU}}$. The derived program $d$ related to $g$, should satisfy $d(a, b) = (b, a)$ which does not require additional operation. Therefore, from Assumption 1 we can deduce that $\mathrm{cost}(g) = 1$ and $\mathrm{cost}(gd) = 1$. We get the same result for $\mathrm{cost}(+)$ by applying identical reasoning.*

**Case 2** ($\mathrm{cost}(\times c), \mathrm{cost}(+c)$) *Let us define $g(a) = c \times a$. To evaluate $g$, we need one operation from $\mathcal{D}_{\mathrm{ReLU}}$. The derived program $d$ related to $g$, should satisfy $d(a) = c$ which does not require additional operation from $\mathcal{D}'_{\mathrm{ReLU}}$. Therefore, from Assumption 1 we can deduce that $\mathrm{cost}(g) = 1$ and $\mathrm{cost}(gd) = 1$. We get the same result for $\mathrm{cost}(+c)$ by applying identical reasoning.*

**Case 3** ($\mathrm{cost}(\log)$) *Let us define $g(a) = \log(a)$. To evaluate $g$, we need one operation from $\mathcal{D}_{\mathrm{ReLU}}$. The derived program $d$ related to $g$, should satisfy $d(a) = 1/a$, which requires the inverse operation from $\mathcal{D}'_{\mathrm{ReLU}}$. Therefore, from Assumption 1 we can deduce that $\mathrm{cost}(g) = 1$ and $\mathrm{cost}(gd) = 2$.*

**Case 4** ($\mathrm{cost}(\exp)$) *Let us define $g(a) = \exp(a)$. To evaluate $g$, we need one operation from $\mathcal{D}_{\mathrm{ReLU}}$. The derived program $d$ related to $g$, should satisfy $d(a) = g(a)$ which does not require operation from $\mathcal{D}'_{\mathrm{ReLU}}$. Finally, from Assumption 1 we can deduce that $\mathrm{cost}(g) = 1$ and $\mathrm{cost}(gd) = 1$.*

**Case 5** ($\mathrm{cost}(inv)$) *Let us define $g(a) = \frac{1}{a}$. To evaluate $g$, we need one operation from $\mathcal{D}_{\mathrm{ReLU}}$. The derived program $d$ related to $g$, should satisfy $d(a) = \frac{-1}{a^2}$ which requires one additional multiplication to compute the square and one $(-1)$ multiplication operation from $\mathcal{D}'_{\mathrm{ReLU}}$. Finally, from Assumption 1 we can deduce that $\mathrm{cost}(g) = 1$ and $\mathrm{cost}(gd) = 3$.*

**Case 6** ($\mathrm{cost}(\mathrm{ReLU})$) *Let us define $g(x) = \mathrm{ReLU}(x) = max(x, 0)$. To evaluate $g$, we need to evaluate the sign of $x$. The derived program $\mathrm{ReLU}'$ can be computed also from the sign of $x$ without further operation. We have $\mathrm{cost}(g) = 1$ by hypothesis, but it is also reasonable to consider $\mathrm{cost}(gd) = 1$ as both operations only require sign evaluation of the same object.*

**Remark 6** Since $\mathcal{D}_{\mathrm{ReLU}}$ dictionary contains the ReLU function, we can build other non-smooth functions such as the maximum and the absolute value. For example, $\max\{x, y\} = \mathrm{ReLU}(x - y) + y = \mathrm{ReLU}(x - y) + \mathrm{ReLU}(y) - \mathrm{ReLU}(-y)$.

### B.2 An extension of Table 1

The justifications of the following are similar to Section B.1, simply taking into consideration different types of operations. Taking $c_{\mathrm{nonlin}} = c_{\mathrm{ReLU}} = 1$, we recover table 1. We replace ReLU by $\times \mathrm{ReLU}$ which corresponds to its usage in practice and allows us to balance the cost of ReLU operations and that of multiplications.

The justification is the same as in Section B.1 taking into consideration different types of operations. For the $\times \mathrm{ReLU}$ operation, the justification is as follows.

**Case 7** ($\times \mathrm{cost}(\mathrm{ReLU})$) *The operation has two argument and requires one sign evaluation and one multiplication in the worst case, so we assign it the cost $1 + c_{\mathrm{ReLU}}$. The differentiated program $d$ should compute the function $(a, b) \mapsto (\mathrm{ReLU}(b), a \times \mathrm{ReLU}'(b))$. One can write a program to compute jointly $g$ and $d$ as follows: return $(a \times b, b, a)$ if $b \geqslant 0$ and $(0, 0, 0)$ if $b < 0$. This only requires a bit sign check which cost is $c_{\mathrm{ReLU}}$ and a multiplication. We therefore model this operation such that $\mathrm{cost}(gd) = \mathrm{cost}(g) = 1 + c_{\mathrm{ReLU}}$.*

Further refinements could be considered including various type of computational operations, such as memory moves, these are beyond the scope of the present paper.

### B.3 Additional elementary nonsmooth programs and cost examples

For simplicity, we do not discuss the dictionary and its related derived dictionary as there are many possibilities, one of them being $\mathcal{D}_{\mathrm{ReLU}}$ and $\mathcal{D}'_{\mathrm{ReLU}}$ as all the considered operations can be equivalently

Table 2: Extension of cost table. $c_{\mathrm{nonlin}} \geqslant 1$ is the cost of nonlinear operations and $c_{\mathrm{ReLU}} \geqslant 0$ is the cost of sign evaluation for ReLU or ReLU$'$.

| $g$ | $(+, \times)$ | $(+c, \times c)$ | log | exp | inv | $\times \mathrm{ReLU}$ |
|---|---|---|---|---|---|---|
| $\mathrm{cost}(g)$ | 1 | 1 | $c_{\mathrm{nonlin}}$ | $c_{\mathrm{nonlin}}$ | $c_{\mathrm{nonlin}}$ | $1 + c_{\mathrm{ReLU}}$ |
| $|\mathtt{pr}|$ | 2 | 1 | 1 | 1 | 1 | 2 |
| $\mathrm{cost}(gd)$ | 1 | 1 | $2c_{\mathrm{nonlin}}$ | $c_{\mathrm{nonlin}}$ | $c_{\mathrm{nonlin}} + 2$ | $1 + c_{\mathrm{ReLU}}$ |
| $\dfrac{\mathrm{cost}(gd)}{\mathrm{cost}(g)}$ | 1 | 1 | 2 | 1 | $\frac{c_{\mathrm{nonlin}}+2}{c_{\mathrm{nonlin}}}$ | 1 |
| $\dfrac{\mathrm{cost}(\times)|\mathtt{pr}|}{\mathrm{cost}(g)}$ | 4 | 2 | $\frac{1}{c_{\mathrm{nonlin}}}$ | $\frac{1}{c_{\mathrm{nonlin}}}$ | $\frac{1}{c_{\mathrm{nonlin}}}$ | $\frac{2}{1+c_{\mathrm{ReLU}}}$ |
| $\dfrac{\mathrm{cost}(gd) + 2\mathrm{cost}(\times)|\mathtt{pr}|}{\mathrm{cost}(g)}$ | 5 | 3 | $\leqslant 4$ | $\leqslant 3$ | $\leqslant 5$ | $\leqslant 5$ |

expressed with ReLU. We use the same framework as in B.2 and we identify the cost of comparing two real numbers with $c_{\mathrm{ReLU}} > 0$. For each program $g$ and associated derived program $d$, we let

$$\omega = \frac{\mathrm{cost}(gd) + 2\mathrm{cost}(\times)|\mathtt{pr}|}{\mathrm{cost}(g)}$$

Table 3: Extension of cost table. $c_{\mathrm{nonlin}} \geqslant 1$ is the cost of nonlinear operations and $c_{\mathrm{ReLU}} \geqslant 0$ is the cost of sign evaluation for ReLU or ReLU$'$. For simplicity $c_{\mathrm{ReLU}}$ is abbreviated $c_{\mathrm{R}}$ and $c_{\mathrm{nonlin}}$ is abbreviated $c_{\mathrm{nl}}$

| $g$ | $(+, \times)$ | $|\cdot|$ | ELU | $3 \times 3$-max-pool | $\|\cdot\|_\infty$ | $\|\cdot\|_1$ |
|---|---|---|---|---|---|---|
| $\mathrm{cost}(g)$ | 1 | $1 + c_{\mathrm{R}}$ | $2 + c_{\mathrm{R}} + c_{\mathrm{nl}}$ | $153 + 8c_{\mathrm{R}}$ | $n + 2nc_{\mathrm{R}} - 1$ | $n(2 + c_{\mathrm{R}}) - 1$ |
| $|\mathtt{pr}|$ | 2 | 1 | 1 | 9 | $n$ | $n$ |
| $\mathrm{cost}(d, g)$ | 1 | $1 + c_{\mathrm{R}}$ | $2 + c_{\mathrm{R}} + c_{\mathrm{nl}}$ | $153 + 8c_{\mathrm{R}}$ | $n + 2nc_{\mathrm{R}} - 1$ | $n(2 + c_{\mathrm{R}}) - 1$ |
| $\dfrac{\mathrm{cost}(gd)}{\mathrm{cost}(g)}$ | 1 | 1 | 1 | 1 | 1 | 1 |
| $\dfrac{\mathrm{cost}(\times)|\mathtt{pr}|}{\mathrm{cost}(g)}$ | 4 | $\frac{1}{1+c_{\mathrm{R}}}$ | $\frac{1}{2+c_{\mathrm{R}}+c_{\mathrm{nl}}}$ | $\frac{9}{153+8c_{\mathrm{R}}}$ | $\frac{n}{n+2nc_{\mathrm{R}}-1}$ | $\frac{n}{n(2+c_{\mathrm{R}})-1}$ |
| $\omega$ | 5 | $\leqslant 3$ | $\leqslant 2$ | $\leqslant 1.12$ | $\leqslant 3$ | $\leqslant 2$ |

**Case 8 (Absolute value and Leaky-ReLU)** *Recall that $|x| = x$ if $x > 0$ and $-x$ otherwise. Similarly Leaky-$\mathrm{ReLU}(x) = x$ if $x > 0$ and $ax$ otherwise, for some parameter $a \in (0, 1)$ so that both cases are exactly the same. The reasoning and result are exactly the same for both operations so we treat the absolute value. The construction is similar as what was proposed for $\times\mathrm{cost}(\mathrm{ReLU})$ treated in the previous section.*

*Let $g$ be a program to evaluate $|\cdot|$, in the worst case it requires one sign evaluation and one multiplication so that $\mathrm{cost}(g) = 1 + c_{\mathrm{ReLU}}$. Similarly it is possible to built a program which returns $(x, 1)$ if $x > 0$ and $(-x, -1)$ otherwise, this computes $(gd)$ and require the exact same operations so that $\mathrm{cost}(gd) = \mathrm{cost}(g) = 1 + c_{\mathrm{ReLU}}$.*

**Case 9 (ELU)**

$$f(x) = \begin{cases} x & \text{if} \quad x \geqslant 0 \\ a(e^x - 1) & \text{if} \quad x < 0 \end{cases} \quad \text{with } a > 0.$$

*Let $g$ be a program to evaluate the ELU function, it requires a sign evaluation and in the worst case one nonlinear operation to evaluate $e^x$, one multiplication to evaluate $ae^x$, and one substraction to evaluate $ae^x - a$. Therefore, $\mathrm{cost}(g) = c_{\mathrm{ReLU}} + c_{\mathrm{nonlin}} + 2$. The derived program $d$ requires the same sign and returns $1$ or $ae^x$ depending on the sign. This does not require additional operation and therefore the joint computation of $g$ and $d$ satisfies $\mathrm{cost}(gd) = \mathrm{cost}(g)$.*

**Case 10 (max-$m$-linear)** *Set $n$ a number of inputs and $m \geqslant 2$ a number of linear functions which are parameters, represented by a matrix $A$ and a fixed input vector of size $n$ represented by $x \in \mathbb{R}^n$. Setting $\max_m \colon \mathbb{R}^m$ to $\mathbb{R}$ the function which evaluates the maximum of $m$ numbers, we consider $g$ a program which evaluates the function $A \mapsto \max_m(Ax)$. Recall that $x$ is fixed so that the number of inputs is $m \times n$. The multiplication requires $m \times (2n - 1)$ multiplications and additions and the evaluation of $\max_m$ requires $(m-1)c_{\text{ReLU}}$ as it requires $m - 1$ pairwise comparisons. We therefore have $\mathrm{cost}(g) = m \times (2n - 1) + (m-1)c_{\text{ReLU}}$.*

*As for the derived program $d$, setting $M_i = 0$ except for row number $i$ which attains the maximum in $g$ which is set to $x$, we have an element of a conservative gradient for $g$. It is possible to jointly compute $g(A)$ and $d(A)$ by invoking a program which returns $((Ax)[i], M_i)$ where $i$ is any index realizing the max and $M_i$ is as discussed. This does not require more operations and we have therefore $\mathrm{cost}(gd) = \mathrm{cost}(g) = m \times 2n - 1 + (m-1)c_{\text{ReLU}}$*

**Case 11 (Two dimensional max-pooling ($3 \times 3$-max-pool))** *We consider a kernel of size $3 \times 3$ for simplicity. The goal is to differentiate with respect to the kernel weights for a fixed input. Let $g$ denote a program implementing such a function, it is of the same form as max-$m$-linear except that the matrix $A$ is of size $9 \times 25$ (padding values at the boundary of the $3 \times 3$ patch, this gives $5 \times 5 = 25$ inputs and $9$ outputs), but it is sparse and can be parametrized by only $9$ values, and the evaluation of the linear function for a fixed $5 \times 5$ input only requires $9 \times (9 + 8) = 153$ addition and multiplications. We then take the maximum of these $9$ outputs so that and $\mathrm{cost}(g) = 153 + 8c_{\text{nonlin}}$. For the same reason as max-$m$-linear, we have $\mathrm{cost}(gd) = \mathrm{cost}(g) = 153 + 8c_{\text{nonlin}}$.*

**Case 12 ($l_1$-norm, $\|\cdot\|_\infty$)** *Denote by $g$ a program which evaluate the $l_1$ norm on $\mathbb{R}^n$. It has $n$ inputs. In the worst case, its evaluation can be done with $n - 1$ addition, $n$ multiplication by $-1$ and $n$ pairwise comparisons. Therefore we have $\mathrm{cost}(g) = 2n + nc_{\text{ReLU}} - 1$. For the same reasons as all examples before, it is possible to identify a derived program $d$ without requiring additional operation so that $\mathrm{cost}(gd) = \mathrm{cost}(g) = 2n + nc_{\text{ReLU}} - 1$.*

**Case 13 (Median of $n$ numbers)** *Denote by $g$ a program that evaluates the median of $n$ numbers. This can be done by sorting the $n$ numbers and outputting the value corresponding to $\lfloor \frac{n}{2} \rfloor$, which requires roughly $n \log(n)$ operations, depending on the algorithm used. The sorting operation is a permutation, one could apply the same permutation to the vector $(1, 2, \ldots, n)$ without additional operation required. The number at position $\lfloor \frac{n}{2} \rfloor$, call it $i$, is the index of the value associated with the median. Setting $d$ to be the null vector in $\mathbb{R}^n$ with value $1$ at position $i$ only, we have a selection in a conservative gradient for the median with no additional operation required. Therefore in this case $\mathrm{cost}(g) = \mathrm{cost}(gd)$.*

**Case 14 (Selection functions)** *This example encompasses virtually all examples used in machine learning and extends the median example above. Assume that $f \colon \mathbb{R}^p \to \mathbb{R}$ is locally Lipschitz, given in the form*

$$f(x) = f_{s(x)}(x)$$

*where $s \colon \mathbb{R}^p \to \{1, \ldots, m\}$ is an index selection function, and for each $i = 1, \ldots, m$, $f_i \colon \mathbb{R}^p \to \mathbb{R}$ is a $C^2$ function. Let $g$ be a program computing $f$, one possibility is to first evaluate $s(x)$ at cost $c_s$ and then evaluate $f_{s(x)}(x)$ at cost $c_f$. As shown in Bolte and Pauwels (2020b), under very mild restrictions on $s$ and $f$ (which should be expressed with logarithms, polynomials, exponentials etc ...), the function*

$$x \mapsto \nabla_{s(x)} f(x)$$

*is a conservative gradient for $f$. It can be seen that it is possible to evaluate jointly $(g, gd)$ by first computing $s$, at a cost $c_s$, then evaluate $f_s$ and $\nabla f_s$ jointly at a cost $c_\nabla$.*

$$\mathrm{cost}(g) = c_s + c_f$$
$$\mathrm{cost}(gd) = c_s + c_\nabla$$
$$\frac{\mathrm{cost}(gd)}{\mathrm{cost}(g)} = \frac{c_s + c_\nabla}{c_s + c_f} \leqslant \frac{c_s + 5c_f}{c_s + c_f}$$

*where we used $c_\nabla \leqslant 5c_f$, the cheap gradient principle for smooth programs. This ratio is close to $5$ if $c_s$ is negligible, we recover the usual ratio for smooth programs. It is close to $1$ if $c_s$ dominates, which is the case in the median example where $f_s$ just corresponds to coordinate number $s$ of the input and has a constant derivative.*

## C    PROOFS OF SECTION 5.1

### C.1    PROOF OF THE MAIN RESULT

**Proof of Theorem 3:**  Let $U \in \mathbb{R}^{p \times p}$ be an orthogonal matrix with entries in $\{-1, 1\}$ which columns are denoted by $u_1, \ldots, u_p$ (with squared norm $p$). Assume that we have as variables a matrix $M \in \mathbb{R}^{p \times p}$ and two matrices $A, B \in \mathbb{R}^{p \times p}$ with columns $a_1, \ldots, a_p$ and $b_1, \ldots, b_p$ respectively.

Consider the function

$$F \colon (x, B, M) \mapsto \frac{1}{p} \sum_{i=1}^{p} |[UB^T Mx]_i|.$$

The pair $(M, B)$ will be identified as $y$ in the statement of the theorem. Considering the dictionary of elementary functions $\{+, \times, \mathrm{ReLU}, +c, \times c\}$, $F$ has a representation as a program $P_F$ using the identity $|t| = \mathrm{ReLU}(t) + \mathrm{ReLU}(-t)$ for all $t \in \mathbb{R}$. We may construct $P_F$ such that $\mathrm{cost}(P_F) = 6p^2 + 2p \leqslant 8p^2$ where we count $2p^2 - p$ operation for each matrix vector multiplication to evaluate $UB^T Mx$ (there are three of them), $p$ multiplication by $-1$ to evaluate $-UB^T Mx$ , $2p$ application of ReLU (on $UB^T Mx$ and $-UB^T Mx$), $p$ additions of ReLU outputs to evaluate $p$ applications of the absolute value, $p - 1$ for the outer sum and 1 for the division. Now consider the constraints

$$\mathrm{sign}(UB^T Ma_i) = u_i, \quad i = 1, \ldots p. \tag{10}$$

The set of matrices $A, B, M$ satisfying this constraint is an open set, call it $S$. We now restrict our attention to this open set and argue that $\mathrm{cost}(P')$ does not change if the input variables are constrained to be in $S$.

We have for all $i = 1, \ldots, p$ and $(A, B, M) \in S$, the following directional derivatives with respect to variable $x$

$$F_1'(0, B, M, a_i) = \frac{1}{p}\mathrm{sign}(UB^T Ma_i)^T UB^T Ma_i = \frac{1}{p} u_i^T UB^T Ma_i = b_i^T Ma_i.$$

Setting the function $G \colon (A, B, M) \mapsto \sum_{i=1}^{p} F_1'(0, B, M, a_i) = \mathrm{Tr}(MAB^T)$, we have that $G$ is a polynomial and $\nabla_M G(A, B, M) = \sum_{i=1}^{p} b_i a_i^T = BA^T$. Note that this does not depend on $M$.

Fix $P'$ any program implementing the directional derivatives function $(y, A) \mapsto F_1'(0, y, A)$ of $F$ described above, with dictionary $\{+, \times, \mathrm{ReLU}, \mathrm{ReLU}', +c, \times c\}$, as in the statement of the theorem.

**Claim 1** *There is a program $P_2$ on dictionary $\mathcal{D} = \{+, \times, +c, \times c\}$ such that $G = [P_2]$ (on the whole space) and $\mathrm{cost}(P_2) \leqslant \mathrm{cost}(P') + p$.*

We use the DAG representation of programs as in Remark 4. Therefore $P'$ is described by a DAG which node are either input nodes or computation nodes implementing functions from $\mathcal{D}'_{\mathrm{ReLU}}$. We will modify the program by simple modifications of the computation nodes. We may obtain a program $P_0$ implementing $G$ on $S$ with dictionary $\mathcal{D}'_{\mathrm{ReLU}}$ with $\mathrm{cost}(P_0) \leqslant \mathrm{cost}(P') + p$ by summing the outputs of $P'$. The $\mathrm{ReLU}'$ nodes in $P_0$ represent a semialgebraic function Coste (2000a;b) with values in a finite set. Therefore, there is a dense open semialgebraic set on which all $\mathrm{ReLU}'$ nodes in $P_0$ are locally constant (Coste, 2000a, Theorem 6.7). Reducing $S$ if necessary, we obtain a program $P_1$ on dictionary $\mathcal{D}_{\mathrm{ReLU}}$ such that $P_1 \sim P_0$ on $S$ by replacing each $\mathrm{ReLU}'$ node in $P_0$ by the corresponding constants. We have $\mathrm{cost}(P_1) \leqslant \mathrm{cost}(P_0)$ (we replace computing nodes by constants). By Lemma 1, there is a program $P_2$ on $\mathcal{D}$ such that $\mathrm{cost}(P_2) = \mathrm{cost}(P_1) \leqslant \mathrm{cost}(P_0) \leqslant \mathrm{cost}(P') + p$ and $G = [P_2]$ (on the whole space). This proves the claim.

We may obtain a program $D_2$ implementing $\nabla_M G$ with dictionary $\mathcal{D}$ by backward algorithmic differentiation on $P_2$, that is $D_2 = \mathrm{backprop}(P_2)$. we have therefore

$$\begin{aligned}
\mathrm{comp}(BA^T, \mathcal{D}) &\leqslant \mathrm{cost}(D_2) \\
&\leqslant \mathrm{cost}(P_2, D_2) \\
&\leqslant 5\mathrm{cost}(P_2) \\
&\leqslant 5p + 5\mathrm{cost}(P'),
\end{aligned}$$

where the first inequality is because $D_2$ is a program computing $BA^T$ for all $A$, $B$ on dictionary $\mathcal{D}$, the second is because adding computation increases the cost, the third is a property of backward algorithmic differentiation on $\mathcal{D}$ and the last one is by construction of $P_2$. Note that $\text{comp}(BA^T, \mathcal{D}) = c(p)$ by definition, therefore we have the claimed lower bound

$$\frac{\text{cost}(P')}{\text{cost}(P_F)} \geqslant \frac{c(p) - 5p}{5\text{cost}(P_F)} = \frac{c(p) - 5p}{8p^2}.$$

$\square$

## C.2 An additional Lemma

**Lemma 1** *Let $Q \colon \mathbb{R}^p \to \mathbb{R}$ be a polynomial and $P_1$ be a program (without loss of generality of level 1) on the dictionary $\mathcal{D}_1 = \{+, \times, \text{ReLU}, +c, \times c\}$, such that $Q = [P_1]$ for all inputs restricted to an open set $S \subset \mathbb{R}^p$. Then there is a level 1 program $P_2$ on the dictionary $\mathcal{D} = \mathcal{D}_1 \backslash \{\text{ReLU}\} = \{+, \times, +c, \times c\}$ such that $Q = [P_2]$ (for all inputs in $\mathbb{R}^p$). Furthermore, if $\text{cost}(\text{ReLU}) = \text{cost}(\times c)$, then, $\text{cost}(P_2) = \text{cost}(P_1)$.*

**Proof :** We use the DAG representation of programs as in Remark 4. Therefore $P_1$ is described by a DAG which node are either input nodes or computation nodes implementing functions from $\mathcal{D}_1$. The function computed by $P_1$ as well as each of its nodes are semi-algebraic Bochnak et al. (2013); Coste (2000a;b). For each ReLU node in the graph representing $P_1$ (assume that there are $N$ of them) we associate a number: the function $\text{ReLU}'$ evaluated on its input (with the convention that $\text{ReLU}'(0) = 0$). This defines a semialgebraic function $G \colon \mathbb{R}^p \to \{0, 1\}^N$. As it has values in a finite set, by semialgebraicity, there is an open subset of $S' \subset S$ such that $G$ is constant on $S$ (Coste, 2000a, Theorem 6.7). Consider $P_2$ which computation graph is the same as that of $P_1$ except that each absolute value node is replaced by multiplication by the corresponding $\text{ReLU}'$ value (which is constant on $S'$). Then $Q = [P_1] = [P_2]$ for all inputs in the open set $S'$. All computation nodes of programs on $\mathcal{D}$ are multivariate polynomials and two polynomials which agree on an open set are equal globally. This concludes the proof. $\square$

# D Proofs of Section 5.3

We investigate in this section the hardness of finding a Clarke subgradient for programs defined on the elementary dictionary $\mathcal{D}_0 = \{+, -, \text{ReLU}\}$. We start with an equivalent representation of these programs as linear ReLU networks with skip connections and specific weight matrices. This equivalence preserve representation size up to polynomial factors. We will then prove a hardness result on such ReLU networks. This will provide proof arguments for Theorem 4 by the polynomial time equivalence of the two representation. We proceed similarly to prove Proposition 1, using the equivalence with the two representations.

## D.1 Polynomial time equivalence with linear ReLU networks with skip connections

Given a set of matrices $M_1 \in \{-1, 0, 1\}^{p_1 \times p}$, $M_2 \in \{-1, 0, 1\}^{p_2 \times p_1}$, $\ldots M_{L-1} \in \{-1, 0, 1\}^{p_{L-1} \times p_{L-2}}$, $M_L \in \{-1, 0, 1\}^{1 \times p_{L-1}}$ we consider the function $F \colon \mathbb{R}^p \to \mathbb{R}$,

$$F \colon x \mapsto M_L \Phi_{L-1}(M_{L-1} \Phi_{L-2}(\ldots \Phi_1(M_1 x))). \tag{11}$$

where $\Phi_i \colon \mathbb{R}^{p_i} \to \mathbb{R}^{p_i}$ are given functions which apply to each coordinate, an activation function which is either the identity or the ReLU function. There is an obvious notion of size for this representation, corresponding to the number of free parameters (matrix entries and coordinates on which ReLU or identity is applied), the size of the representation is $p_{L-1} + \sum_{i=1}^{L-1} p_i \times p_{i-1} + p_i$.

A function $F$ given in (11) can be represented by a program on $\mathcal{D}_0$ of equivalent size, this correspond to a naive implementation. Similarly, any program $P \in \mathcal{P}(\mathcal{D}_0)$ on $p$ inputs and with a single output can be represented by a network as in (11) which size is at most $18\text{cost}(P)^3$. Indeed, we may assume that $\text{cost}(P) \geqslant p/2$ without loss of generality, otherwise, the program would not perform operations on some of the input variables and it could be simplified by removing variables which do not affect the output. Recall that $m$ in Algorithm 1 is the memory footprint of $P$, in our case, it

is $m = p + \text{cost}(P)$, the number of inputs plus the total number of operations. Note that we have $m \leqslant 3\text{cost}(P)$. Each operation $+, -$ or ReLU in the program can be represented by a $m \times m$ matrix composed with a certain $\Phi \colon \mathbb{R}^m \to \mathbb{R}^m$ which contribution to the Relu network size is at most $(m^2 + m) \leqslant 2m^2 \leqslant 18\text{cost}(P)^2$ since $m$ is integer and $m \leqslant 3\text{cost}(P)$. There are $\text{cost}(P)$ such operations so that a program can be represented equivalently by linear Relu network, with $L = \text{cost}(P)$ layers which contribution to the network size is at most $18\text{cost}(P)^2$ so that the size of the resulting network is at most $18\text{cost}(P)^3$, which is the desired bound since.

We have shown that working with functions represented as in equation (11) is equivalent to work with programs in $\mathcal{P}(\mathcal{D}_0)$ as it is possible to switch from one to the other at a cost of an increase of the representation size which is only cubic. Therefore we will from now on work with functions represented as linear relu networks with skip connections as in (11), and NP-hardness or polynomial time results on such function will be valid on $\mathcal{P}(\mathcal{D}_0)$ by the construction above.

### D.2 FURTHER PROPERTIES OF LINEAR ReLU NETWORKS

Throughout this section $F$ denotes a with representation as in (11). This function is positively homogeneous, it satisfies $F(0) = 0$ and it. By piecewise linearity, its Clarke subdifferential is a polyhedron (see e.g., Arora et al. (2018); Raghu et al. (2017)). The Clarke subdifferential is a conservative gradient for this function, and we will associate to it a different conservative gradient, associated to Algorithm 2

**Definition 2 (Autodiff conservative gradient)** We consider a specific conservative gradient for $F$, it is given by $D_F^a(x) = \{M_1^T D_1 M_2^T D_2 \ldots M_{L-1}^T D_{L-1} M_L^T\}$, where for $i = 1, \ldots, L-1$, $D_i$ is a diagonal matrix which entries respects the sign pattern of the corresponding activation function: 1 if the activation is identity, 0 if the activation is ReLU and the input is negative, 1 if the input is positive and all elements in $[0, 1]$ if the input is null. We have in particular

$$D_F^a(0) = \{M_1^T D_1 M_2^T D_2 \ldots M_{L-1}^T D_{L-1} M_L^T\} \tag{12}$$

where in this case, diagonal entries of matrices $D_i$ corresponding to ReLU activations are arbitrary in $[0, 1]$ and the remaining diagonal entries are 1 (corresponding to identity activations).

The autodiff conservative gradient is associated with the algorithmic differentiation of a natural numerical program implementing $F$ as in Subsection 3.2. Furthermore, one can check that given a program $P \in \mathcal{P}(\mathcal{D}_0)$, after the transformation outlined in Section D.1, we have that $D_F^\alpha$ coincides with $D_P$ in Theorem 2. In the following definition, $D_F$ could be, for example, the Clarke subdifferential of $F$ or the algorithmic differentiation conservative gradient $D_F^a$.

We consider the following problem.

**Problem 1 (Conservative gradient enumeration)** Given matrices $M_1 \in \mathbb{R}^{p_1 \times p}$, $M_2 \in \mathbb{R}^{p_2 \times p_1}$, $\ldots M_{L-1} \in \mathbb{R}^{p_{L-1} \times p_{L-2}}$, $M_L \in \mathbb{R}^{1 \times p_{L-1}}$, and functions $\Phi_1, \ldots, \Phi_{L-1}$, consider $F \colon \mathbb{R}^p \to \mathbb{R}$ the associated linear ReLU network with skip connections in (11), $x \in \mathbb{R}^p$ and $D_F \colon \mathbb{R}^p \rightrightarrows \mathbb{R}^p$ a conservative gradient for $F$. Compute two distinct elements in $D_F(x)$ or one element if it is a singleton.

This problem enters the field of computational complexity as we have associated to it a representation size corresponding to the number of "free parameters" to be chosen: each matrix entry and the activation (ReLU or identity) corresponding to each coordinate, resulting in a number of parameters $p_{L-1} + \sum_{i=1}^{L-1} p_i \times p_{i-1} + p_i$. In what follows, we will consider integral or rational entries for matrices and input $x$ with the common notion of bit size. Schrijver (1998).

#### D.2.1 CLARKE ENUMERATION IS NP-HARD FOR ReLU NETWORKS

The decision version of Problem 1, under the same assumptions, is to decide if there exists two distinct elements in $D_F(x)$, that is, decide if $D_F(x)$ is not reduced to a singleton.

**Theorem 5 (Finding two Clarke subgradients is NP-Hard)** *Decision version of problem* (1) *with matrix and vector entries in* $\{-1, 0, 1\}$ *and* $D_F = \partial^c F$ *is NP-hard.*

**Sketch of proof:** We encode a boolean formula $\pi$ on $p$ boolean variable, in a linear ReLU network with $p$ inputs, of size proportional to that of $\pi$. We do so by replacing "or" operations by maxima, "and" operations by minima, negation by multiplication by $-1$ and adding ReLU operations to the result. Using Lemma 3 in appendix D.5, the resulting $F$ is represented by a linear ReLU network. By construction, 0 is a global minimum of $F$ so $0 \in \partial^c F(0)$, and $F$ takes positive values if and only if $\pi$ is satisfiable if and only if $\partial^c F(0) \neq \{0\}$. We detail this proof in coming sections.

Theorem 5 illustrates the hardness enumerating Clarke subgradients of linear ReLU networks. For $F$ as in (11) and $x \in \mathbb{R}^p$, $\partial^c F(x)$ is not a singleton if and only if $F$ is not differentiable at $x$, therefore:

**Corollary 2 (Deciding non-differentiability of a NN is NP-Hard)** *Given a linear ReLU network as in* (11) *with matrices as in Theorem 5 and* $x \in \mathbb{R}^p$*, deciding if* $F$ *is not differentiable at* $x$ *is NP-hard.*

In the coming section, we will provide a proof for Theorem 5 and Corollary 2. By the polynomial time equivalence of the representation of programs in $\mathcal{P}(\mathcal{D}_0)$ and functions as in (11) detailed in Section D.1, this proves Theorem 4.

We add a remark on lexicographic subdifferential. It follows from (Barton et al., 2018, Proposition 2.7) that, for linear ReLU network $F$ as in (11), the lexicographic subdifferential Nesterov (2005) is the set of neighboring gradients and is contained in Clarke subdifferential.

**Corollary 3 (Finding two lexicographic subgradients is NP-Hard)** *Theorem 5 remains true if* $D_F$ *is the lexicographic subdifferential.*

### D.3 PROOF OF THE MAIN HARDNESS RESULT

**Preliminary on 3-SAT** We will use reduction to 3-SAT problem which is among the most well known NP-complete problems. Recall that a boolean formula is built from boolean variables, and operators: AND (conjunction, denoted $\wedge$) OR (disjunction, $\vee$) and NOT (negation, $\neg$). A literal, is either a variable or the negation of a variable. A clause is a disjunction of literals (or a single literal). A formula is in conjunctive normal form (CNF), if it is a conjunction of clauses or a clause. 3-SAT is the decidability problem associated to CNF formulas with clauses containing 3 literals, such formulas are called 3-CNF formulas.

**Example 2** The formula $(b_1 \vee b_2 \vee \neg b_3) \wedge (b_1 \vee b_4 \vee \neg b_5) \wedge (\neg b_2 \vee \neg b_3 \vee b_6)$ is 3-CNF with 6 boolean variables $b_1, \ldots, b_6$ and 3 clauses.

**Problem 2 (3-SAT)** *Given* $p, n \in \mathbb{N}$ *and a boolean function* $\pi$ *with* $p$ *boolean arguments* $b_1, \ldots, b_p$ *represented by a 3-CNF formula with* $n$ *clauses, decide if there exists an assignment* $(b_1, \ldots, b_p) \in \{0, 1\}^p$ *such that* $\pi(b_1, \ldots, b_p) = 1$.

**Proof of Theorem 5:**

The reduction is to 3-SAT.

Consider a 3-CNF function $\pi$ in $p$ variables $b_1, \ldots, b_p$ with $n$ clauses of size 3. We may assume without loss of generality that $n$ is of the form $2^k$ for $k \in \mathbb{N}$ by adding clauses which are always true and increasing the number of clauses by a factor at most 2. We will consider $p$ real variables $x_1, \ldots, x_p$. Consider the first clause of $\pi$, say for example $(b_1 \vee b_2 \vee \neg b_3)$. We associate to each literal the corresponding variable $x$ if the literal is equal to a variable, and $-x$ if it is the negation of the corresponding variable, for example $x_1, x_2, -x_3$. These are combined using $\mathrm{ReLU} \circ \max$ resulting in the expression $\mathrm{ReLU}(\max\{x_1, x_2, -x_3\})$.

We proceed similarly for each clause, we obtain $n = 2^k$ expressions involving $\mathrm{ReLU} \circ \max$ where the $\max$ is over three numbers. The $\max$ of 3 numbers is the same as the $\max$ of 4 numbers (by copying one of the inputs) and, according to Lemma 3, can be represented by a ReLU network with 2 ReLU layers of size at most $3 \times 2 = 6$ with weight matrices in $\{-1, 0, 1\}$.

We may therefore represent the $n$ $\mathrm{ReLU} \circ \max$ expressions with a network with $p$ inputs and $n$ outputs, with 3 ReLU layers (2 for each $\max$ and one for the outer ReLU) of size at most $6n$ (6 nodes for each $\max$) involving matrices with entries in $\{-1, 0, 1\}$. These expressions are

combined using the operator $\min$ applied to the $n = 2^k$ clause. Thanks to Lemma 3 again, using $\min\{a, b\} = -\max\{-a, -b\}$, the max over the $2^k$ numbers can be expressed with $k$ layers of size at most $3 \times 2^{k-1} = \frac{3}{2}n$

We call the resulting network $F$. It has a representation as in (11), with matrices with entries in $Z_3 = \{-1, 0, 1\}$ as in Problem 1. It contains $\log_2(n) + 3$ ReLU layers of size at most $6n$ and it has therefore a description which size is polynomially bounded in $n$ which is proportional to the bit size representation of the 3-CNF formula $\pi$.

**Example 3** If the 3-CNF formula is given by $(b_1 \vee b_2 \vee \neg b_3) \wedge (b_1 \vee b_4 \vee \neg b_5) \wedge (\neg b_2 \vee \neg b_3 \vee b_6) \wedge (b_2 \vee \neg b_2 \vee b_6)$ with $p = 6$ boolean variables and $n = 4$ clauses, we will obtain a network computing the following expression in 6 real variables $x_1, \ldots, x_6$:

$$
\begin{aligned}
&F(x_1, \ldots, x_6) \\
&= \min(\text{ReLU}(\max(x_1, x_2, -x_3)), \text{ReLU}(\max(x_1, x_4, -x_5)), \\
&\qquad \text{ReLU}(\max(-x_2, -x_3, x_6)), \text{ReLU}(\max(x_2, -x_2, x_6)))).
\end{aligned}
$$

We have the following rules for $\min$ and $\max$ over real numbers $a, b, c$ (we use the convention $\text{sign}(0) = 0$).

- $\max(a, b, c) > 0 \qquad \Leftrightarrow \qquad (a > 0) \vee (b > 0) \vee (c > 0)$.
- $\max(a, b, c) > 0 \qquad \Leftrightarrow \qquad \max(\text{sign}(a), \text{sign}(b), \text{sign}(c)) > 0$.
- $\min(a, b, c) > 0 \qquad \Leftrightarrow \qquad (a > 0) \wedge (b > 0) \wedge (c > 0)$.
- $\min(a, b, c) > 0 \qquad \Leftrightarrow \qquad \min(\text{sign}(a), \text{sign}(b), \text{sign}(c)) > 0$.
- $a > 0 \qquad \Leftrightarrow \qquad (-a < 0) \qquad \Leftrightarrow \qquad \text{sign}(a) > 0$.
- $\text{ReLU}(\max(\text{sign}(a), \text{sign}(b), \text{sign}(c))) \in \{0, 1\}$.

Because of the $\min \circ \text{ReLU}$ structure, we have $F(x) \geq 0$ for all $x$, furthermore, $F(0) = 0$, so that $0$ is a global minimum of $F$ and $0 \in \partial^c F(0)$. For any $x$, we have $F(x) > 0$ if and only if the output of each $\max$ is positive, if and only if each $\max$ clause contains a positive argument. We therefore have that $F(x) > 0$ if and only if $F(\text{sign}(x)) > 0$ where $\text{sign}$ is the coordinatewise application of the sign, taking value $0$ at $0$.

We have the following chain of equivalence

$$
\begin{aligned}
&\partial^c F(0) \neq \{0\} \\
\Leftrightarrow \quad &\exists x \in \mathbb{R}^p, \quad F(x) \neq 0 \\
\Leftrightarrow \quad &\exists x \in \mathbb{R}^p, \quad F(x) > 0 \\
\Leftrightarrow \quad &\exists x \in \mathbb{R}^p, \quad x_i \neq 0 \,(\forall i = 1, \ldots, p) \quad F(x) > 0 \\
\Leftrightarrow \quad &\exists x \in \mathbb{R}^p, \quad x_i \neq 0 \,(\forall i = 1, \ldots, p) \quad F(\text{sign}(x)) > 0 \\
\Leftrightarrow \quad &\exists x \in \{-1, 1\}^p, F(x) > 0 \\
\Leftrightarrow \quad &\exists x \in \{-1, 1\}^p, \quad \pi(b) = 1, \quad b_i = \mathbb{I}(x_i = 1) \quad (i = 1 \ldots p),
\end{aligned}
$$

where $\mathbb{I}$ outputs $1$ if the boolean argument is true, and $0$ otherwise. The first equivalence is by Lemma 2, the second is because $F \geq 0$, the third is because $F$ is continuous, the fourth is by the discussion above and the fifth is obvious because all possible $\{-1, 1\}$ patterns can be described as coordinatewise sign applied vectors in $\mathbb{R}^p$ with nonzero entries. For the last equivalence, for $x_i \in \{-1, 1\}$ we set $b_i = 0$ if $x_i = -1$ and $b_i = 1$ if $x_i = 1$. Each $\text{ReLU} \circ \max$ applied to the sign vector corresponds to a clause and its output is in $\{0, 1\}$. The output of each $\text{ReLU} \circ \max$ clause is $1$ if and only if at least one of its argument is $1$, if and only if one of the litteral of the corresponding disjunction is $1$ if and only if the disjunction applied to the corresponding boolean variables is true. Otherwise, it is $0$. Similarly, the $\min$ combination has positive output if and only if all $\max$ outputs are $1$ if and only if all the disjunctions applied to variables $b_i$ are true.

This shows that Problem 1 is NP-hard, because $0 \in \partial^c F(0)$ and $\partial^c F(0) \neq \{0\}$ if and only if there exists two distinct elements in $\partial^c F(0)$. □

### D.4    PROOF OF FEASIBILITY FOR AUTODIFF CONSERVATIVE GRADIENT

The counterpart of Problem 1 for AD conservative gradient in Definition 2 is tractable, illustrating a major computational difference between Clarke subdifferential and AD conservative gradient. The proof is in Section D.4, by reduction to a graph shortest path problem. By the polynomial time equivalence between linear ReLU network and programs on $\{+, -, \mathrm{ReLU}\}$ proved in Section D.1, this proves Proposition 1.

**Proposition 2** *Problem* (1) *with matrix entries in $\mathbb{Q}$ and $D_F = D_F^a$ is polynomial time solvable.*

**Proof of Proposition 2:**  Consider the following polynomial expression:

$$M_1^T(\bar{Q}_1 + Q_1)\ldots M_{L-1}^T(\bar{Q}_{L-1} + Q_{L-1})M_L^T, \tag{13}$$

where we decomposed $D_i = \bar{Q}_i + Q_i$ in Definition 2, such that $\bar{Q}_i$ is constant, diagonal, with zero entries except for the $1$ entries which are enforced by the network activation and sign pattern: strictly positive activation before application of ReLU when network is evaluated at $x$, or identity activations. Furthermore, $Q_i$ contains $q_i \leqslant p_i$ diagonal variables to be chosen in $[0, 1]$ corresponding to the zero activation pattern before application of ReLU, for $i = 1, \ldots, L - 1$. The strictly negative values before application of ReLU do not play an additional role, they correspond diagonal entries constrained to $0$ in both $\bar{Q}_i$ and $Q_i$, $i = 1, \ldots, L - 1$. Note that a polynomial is constant on a box if and only if it is constant so the polynomial expression in (13) is constant when diagonal entries are constrained in $[0, 1]$, if and only if it is constant. So the problem reduces to decide if the polynomial expression in (13) is non constant, with respect to variables $Q_1, \ldots, Q_{L-1}$. We show that this reduces to a graph connectivity problem over $2 + \sum_{i=1}^{l-1} q_i$ vertices and edge weight given by partial products in (13).

First, the problem can be reduced to finding a non-zero value in the expression in (13). Indeed, one can substract the value obtained choosing $Q_i = 0$, $i = 1, \ldots, L - 1$ and use the following block representation:

$$\begin{pmatrix} M_1^T & -M_1^T \end{pmatrix} \begin{pmatrix} \bar{Q}_1 + Q_1 & 0 \\ 0 & \bar{Q}_1 \end{pmatrix} \ldots \begin{pmatrix} M_{L-1}^T & 0 \\ 0 & M_{L-1}^T \end{pmatrix} \begin{pmatrix} \bar{Q}_{L-1} + Q_{L-1} & 0 \\ 0 & \bar{Q}_{L-1} \end{pmatrix} \begin{pmatrix} M_L^T \\ M_L^T \end{pmatrix}$$
$$= M_1^T(\bar{Q}_1 + Q_1)\ldots M_{L-1}^T(\bar{Q}_{L-1} + Q_{L-1})M_L^T \quad - \quad M_1^T\bar{Q}_1\ldots M_{L-1}^T\bar{Q}_{L-1}M_L^T. \tag{14}$$

Therefore, expression (13) is nonconstant if and only if expression in (14) takes a nonzero value for some assignment of $Q_1, \ldots, Q_{L-1}$. The number of variables in (13) and (14) is the same and they have exactly the same form. Therefore we assume without loss of generality that the problem is to decide if the polynomial expression in (13) is not equal to the null polynomial.

Expression (13) is a vector function each of its coordinates being a polynomial function. It is not uniformly null if and only if and only if there exists a coordinate which is not the null polynomial, so we may add a diagonal matrix $Q_0$ with $p_0 = p$ diagonal entries in $[0, 1]$ (and $\bar{Q}_0 = 0$ for the sake of symmetry) and $M_0 \in \mathbb{R}^{p \times 1}$ the vector of all ones and find a nonzero value for the product

$$M_0^T(\bar{Q}_0 + Q_0)M_1^T(\bar{Q}_1 + Q_1)\ldots M_{L-1}^T(\bar{Q}_{L-1} + Q_{L-1})M_L^T, \tag{15}$$

Expression (15) is now real valued and therefore defines a polynomial. For each $0 = 1\ldots L - 1$, denote by $d_i \in [0, 1]^{q_i}$, the vector containing the diagonal entries of matrix $Q_i$, this corresponds exactly to the variable diagonal elements of $D_i$ in Definition 2. Denote by $P(d_0, \ldots, d_L)$ the obtained polynomial, $P$ is multilinear in $d_0, \ldots, d_{L-1}$, that is, it has an affine dependency for one block vector if the others are fixed. Therefore the hessian of $P$ has zero diagonal blocks and the function is harmonic (hessian has zero trace), as a consequence, the maximum principle for harmonic functions entails that its maximum and minimum on any polytope are attained at vertices.

For $i = 0, \ldots, L - 1$ denote by $\Delta_i \subset \mathbb{R}^{q_i}$, the convex hull of the origin and the canonical basis vectors, this is a $q_i$ dimensional simplex with nonempty interior. The polynomial $P$ in (15) is identically zero if and only if it vanishes on the product of simplices $\Delta_0 \times \ldots \times \Delta_{L-1}$ (which has non empty interior), if and only if it vanishes on the product set of the edges of these simplices by the maximum principle. In other words, $P$ is not identically zero, if and only if it contains a nonzero element when each $d_i$ is restricted to be an element of the canonical basis (zero vector with exactly one nonzero entry) or the null vector.

Define a graph with a layer structure:

- The source layer $V_{-1}$ contains a single source node, $v_{-1,1}$.
- The zero-th layer $V_0$ contains $q_0 = p$ nodes $v_{0,1} \ldots v_{0,q_0}$.
- Recursively, the $i$-th layer $V_i$ contains $q_i$ nodes $v_{i,1} \ldots v_{i,q_i}$, for $i = 1 \ldots L - 1$.
- The sink layer $V_L$ contains a single node node $v_{L,1}$.

We connect nodes between consecutive layers, respecting the order induced by the layer structure. For $i = -1, \ldots L - 1$ and $j = 0, \ldots, L$, with $j > i$, we connect layers $V_i$ and $V_j$ as follows

- Compute the quantity

$$R = \left( \prod_{m=i+1}^{j-1} M_m^T \bar{Q}_m \right) \times M_j^T,$$

where if $j = i + 1$ the product reduces to the identity ($R = M_j^T$).
- For $k = 1, \ldots, q_i$ and $l = 1, \ldots, q_j$, add an edge with between $v_{i,k}$ and $v_{j,l}$ if $R_{k,l} \neq 0$.

The resulting graph has a number of nodes equal to the number of ReLU functions in $F$ plus $p$ additional nodes and the source and sink nodes. Computation of edges can be done in polynomial time: it requires at most $4(L + 1)^2$ matrix products involving at most $2L + 1$ matrices. Indeed the product of $m$ matrices has polynomial time complexity in the representation bit size of the $m$ input matrices.

In this graph, a directed path from the source to the sink visits each layer at most once, and in that case it visits a single node. Each such path corresponds to a monomial with nonzero coefficient appearing in the polynomial $P$ in (15) by construction of the graph structure. Conversely each nonzero coefficient of a given monomial in (15) is uniquely associated to a path in the graph which corresponds to the nodes associated to variables in the monomial. Therefore, the source is connected to the sink if and only if there is a nonzero monomial in (15), if and only if the corresponding polynomial is nonzero. Furthermore, each path corresponds to the evaluation of the program at an edge of the product $\Delta_0 \times \ldots \times \Delta_{L-1}$. Therefore finding a path connecting the source to the sink allows to compute a nonzero element in the product and if no such path exists, the polynomial is identically zero.

So we have shown that the truth value of problem 1 with $D_F = D_F^a$, is the same as the source being connected to the sink by a directed path in the graph we defined, which has size polynomialy bounded compared to network size. Connectivity can be solved, for example using Dijkstra's algorithm, in time $O(|V|^2)$ where $|V|$ is the number of nodes (or vertices). A path represents a nonzero element of $D_f(0)$ and if no such path exists, we conclude that $D_F(0) = \{0\}$. This shows that the problem is solvable in polynomial time and concludes the proof.

$\square$

## D.5 ADDITIONAL LEMMAS

The following lemma provides a characterization of singleton subgradient for linear ReLU networks.

**Lemma 2** *Let $F$ be a linear ReLU network, then $\partial^c F(0) = \{0\}$ if and only if $F$ is constant.*

**Proof :** If $F$ is constant, the result is immediate because $F \equiv 0$. Now, suppose that $\partial^c F(0) = \{0\}$. We know that $F$ is piecewise linear and there exists a finite set of polyhedron whose union is $\mathbb{R}^p$, where $F$ is affine linear over each polyhedron. Furthermore, $F$ is positively homogeneous, therefore for each $x \in \mathbb{R}^p$, $\partial^c F(x) = \partial^c F(\lambda x)$ with $\lambda > 0$. Setting $R \subset \mathbb{R}^p$, the full measure set where $F$ is differentiable, one has that for all $x \in \mathbb{R}^p$ and

$$\partial^c F(x) = \operatorname{conv} \left\{ v \in \mathbb{R}^p, \, \exists y_k \underset{k \to \infty}{\to} 0 \text{ with } y_k \in R, \, v_k = \nabla F(y_k) \underset{k \to \infty}{\to} v \right\} = \{0\}.$$

Therefore, each affine part has zero derivative on each polyhedra and by continuity we conclude that $F$ is constant. $\square$

The next lemma describes an explicit representation of maximum of finitely many numbers using a ReLU network with weights in $\{-1, 0, 1\}$.

**Lemma 3** *Given $k \in \mathbb{N}$, $k > 0$, there exists $F$, a ReLU network with $k$ ReLU layers of size at most $3 \times 2^{k-1}$ and weight matrices with entries in $\{-1, 0, 1\}$, with $p = 2^k$ inputs such that for any $x \in \mathbb{R}^p$,*

$$F(x) = \max_{i=1,\ldots,2^k} x_i.$$

**Proof :** We proceed by recursion on $k$. Note that for any $x_1, x_2 \in \mathbb{R}$

$$\max\{x_1, x_2\} = \text{ReLU}(x_1 - x_2) + x_2 = \text{ReLU}(x_1 - x_2) + \text{ReLU}(x_2) - \text{ReLU}(-x_2).$$

Set the matrices

$$A = \begin{pmatrix} 1 & -1 \\ 0 & 1 \\ 0 & -1 \end{pmatrix} \qquad B = \begin{pmatrix} 1 & 1 & -1 \end{pmatrix}.$$

The function $F_1 \colon \mathbb{R}^2 \to \mathbb{R}$ given by

$$F_1(x) = B\text{ReLU}(Ax)$$

satisfies $F_1(x) = \max\{x_1, x_2\}$. This proves the result for $k = 1$.

Now assume that for $k \geqslant 1$, we have a network with $k$ ReLU layers of size at most $3 \times 2^k$ represented by matrices $M_1, \ldots, M_{k+1}$ with entries in $\{-1, 0, 1\}$, such that the corresponding ReLU network, as in (11) $F_k \colon \mathbb{R}^{2^k} \to \mathbb{R}$ satisfies for all $x \in \mathbb{R}^{2^k}$,

$$F_k(x) = \max_{i=1,\ldots,2^k} x_i.$$

Set $\tilde{F}_k$ the concatenation of two copies of $F_k$, that is $\tilde{F}_k \colon \mathbb{R}^{2^{k+1}} \to \mathbb{R}^2$, such that for all $x, y \in \mathbb{R}^{2^k}$,

$$\tilde{F}_k(x, y) = \begin{pmatrix} \max_{i=1,\ldots,2^k} x_i \\ \max_{i=1,\ldots,2^k} y_i \end{pmatrix}.$$

The matrices representing $\tilde{F}_k$ can be described in block form

$$\tilde{M}_i = \begin{pmatrix} M_i & 0 \\ 0 & M_i \end{pmatrix} \in \mathbb{R}^{(2p_i) \times (2p_{i-1})}$$

for $i = 1, \ldots, k+1$, where $p_0 = 2^k$ and $p_k = 1$. This network is made of $k$ layers of size at most $3 \times 2^{k+1}$, it has $2^{k+1}$ inputs and two outputs and its weight matrices have elements in $\{-1, 0, 1\}$. The block representation of the last matrix of this network is of the form

$$\begin{pmatrix} M_{k+1} & 0 \\ 0 & M_{k+1} \end{pmatrix} \in \mathbb{R}^{2 \times l}$$

where $l$ is the size of the row vector $M_{k+1}$. We have

$$A \times \tilde{M}_{k+1}$$
$$= \begin{pmatrix} 1 & -1 \\ 0 & 1 \\ 0 & -1 \end{pmatrix} \times \begin{pmatrix} M_{k+1} & 0 \\ 0 & M_{k+1} \end{pmatrix} = \begin{pmatrix} M_{k+1} & -M_{k+1} \\ 0 & M_{k+1} \\ 0 & -M_{k+1} \end{pmatrix} \in \mathbb{R}^{3 \times (2l)}.$$

We set $F_{k+1}(x, y) = F_1(F_k(x), F_k(y)) = F_1(\tilde{F}_k(x, y))$ for all $x, y \in \mathbb{R}^{2^k}$. In matrix notation we have

$$F_{k+1}(x, y) = B\text{ReLU}(A\tilde{F}_k(x, y)).$$

The involved matrices are $M_{k+2} = B$, $A \times \tilde{M}_{k+1}$ and $\tilde{M}_k \ldots \tilde{M}_1$. They all have entries in $\{-1, 0, 1\}$ and the corresponding network has layers of size at most $3 \times 2^{k+1}$. The result then holds by recursion.
$\square$

