# OpenReview forum: "On the complexity of nonsmooth automatic differentiation"
_ICLR.cc/2023/Conference — ICLR 2023 notable top 25%_

### Official Review · Reviewer_vqhQ · 2022-10-24

**Confidence:** 2
**Correctness:** 4
**Technical Novelty And Significance:** 3
**Empirical Novelty And Significance:** Not applicable
**Recommendation:** 6

**Clarity, Quality, Novelty And Reproducibility:**

The extension to the nonsmooth functions using conservative gradients seems novel.

**Strength And Weaknesses:**

The paper points out the different complexity for computing derivatives for non-smooth functions, and shows a variety of results regarding the complexity ratio of different derivative methods.

- the paper provides and rigorously prove bounds for the computational hardness non-smooth AD as well as for different generalized gradient methods.
- The authors show that calculating distinct elements in a clarke differential of a given point is NP-hard.
- The proofs (as much as i could follow them) seem correct.

However, i think the proof sketches could be made more clearer in the main text, since it was a bit hard to follow.

**Summary Of The Paper:**

This paper utilizes the concept of a conservative gradient to bound the complexity of non-smooth derivatives, and shows that the computational complexity for backprop with conservative gradients has a computational overhead ratio independent of dimension (Theorem 2), and hence generalizes Baur-Strassen to non-smooth programs.

The authors show that in general for other differentiation oracles (like directional derivatives, etc) for the forward mode the overhead depends upon the dimension p (for example is O(p^\omega) for directional derivatives), and therefore can be limited.

**Summary Of The Review:**

The authors extend the cheap gradient principle and show that it is independent of the dimension. They further show that bound the forward auto-diff complexity will have an overhead due to matrix multiplication.

---

> ### Author Response · Authors · 2022-11-10
> **Response to Reviewer vqhQ**
>
> We thank the reviewer for evaluating our submission.
>
> ### However, I think the proof sketches could be made more clearer in the main text, since it was a bit hard to follow.
>
> It is true, we will simplify the sketch of the evidence in the main text.

---

> > ### Comment · Reviewer_vqhQ · 2022-11-20
> > **Thank you for the rebuttal.**
> >
> > Thank you!

---

### Official Review · Reviewer_9fsL · 2022-10-27

**Confidence:** 4
**Correctness:** 4
**Technical Novelty And Significance:** 3
**Empirical Novelty And Significance:** 3
**Recommendation:** 6

**Clarity, Quality, Novelty And Reproducibility:**

The paper is well-written, the proof are clean and the previous works are well-cited.


**Strength And Weaknesses:**

Strengths
======================
  - The problem of theoretically understanding the complexity of computing subgradients of non-smooth functions is very important and interesting.

Weaknesses - Comments
========================

  1. The main problem with the paper is that most of the material is a restatement of results provided by Bolte and Pauwels 2020a. In particular, the algorithms for forward and backward computation of conservative gradients are provided in exactly the same form in Bolte and Pauwels 2020a, at least in their arxiv version. Given that, the novelty of this part of the paper is the proof of Theorem 2, which can be found in page 15 and mostly follows a very simple induction argument similar to the existing proofs for the smoothed case.

  2. I am not sure I understand the motivation of Theorem 3. Why would we need to compute the directional derivatives since we can compute the whole subgradient?

  3. Related to Theorem 4: The problem of given a function $f$, a vector $v$, and a point $x$ verify whether $v$ is a subgradient of $f$ at $x$ is solvable in polynomial time or not?

  4. If the answer to question 3 is positive then I am puzzled with Theorem 4 (b). The authors state that deciding the differentiability of a non-smooth function at a point $x$ is NP-hard. But I believe that the correct statement is that deciding the non-differentiability of $f$ is NP-hard. In fact, if the answer to 3 is positive, deciding the non-differentiability should be NP-complete, since the existence of two subgradients is a witness of the non-differentiability of $f$ at $x$. This means that deciding the differentiability is coNP-complete and we do not expect it to either be in NP or being NP-hard.

  5. There is a small issue with Theorem 4 that usually is swept under the rug, but when proving computational hardness results it needs attention. What is the representation of the numbers that you assume in Theorem 4?
  **(a)** If you use binary representation, then the hardness result of Theorem 4 implies that there is no algorithm with complexity polynomial in $p$ (the dimension) and $\log(1/\epsilon)$, where $\epsilon$ is the accuracy that we choose to represent the numbers.
  **(b)** If you can show the hardness result even when you use unary representation, then the Theorem 4 implies that there is no algorithm with complexity polynomial in $p$ and $1/\epsilon$.
  If the case is (a) then this would be a less interesting result because the ML community focuses on algorithms with running time $\mathrm{poly}(1/\epsilon)$. In any case, a discussion about this is missing and is important.

**Summary Of The Paper:**

  In this paper, the authors consider the problem of computing subgradients of non-smooth non-convex functions that are given as programs that are based on the small set of elementary functions. Their main contributions are the following:
  1. They analyze the computational complexity of the algorithm suggested by Bolte and Pauwels 2020a for forward and backward evaluations of a relaxation of subgradients called conservative gradients. An important corollary of their analysis is that when we apply backward evaluation, the cost of evaluating the gradient is only a dimension-independent constant time larger than the cost of evaluating the function value. The authors also show that this is not the case for the forward evaluation of the gradient.
  2. They show that given a program P that computes a function $f$ using only the operations $\{ +, -, \mathrm{ReLu} \}$, and a point $x$, deciding whether $f$ is differentiable at $x$ is NP-hard. In contrast deciding whether $f$ is path-differentiable at $x$, which means that there is a unique conservative gradient, is possible in polynomial time.

**Summary Of The Review:**

  The contribution of this paper is interesting but I am not sure if it is enough to guarantee acceptance at ICLR. For this reason I am slightly leaning towards rejection.

---

> ### Author Response · Authors · 2022-11-10
> **Response to Reviewer 9fsl (1/2)**
>
> We thank the reviewer for his comments and constructive feedback. Comments are answered in detail below. We believe that they address the concerns of the reviewer.
>
>
> ### The main problem with the paper is that most of the material is a restatement of results provided by Bolte and Pauwels 2020a. In particular, the algorithms for forward and backward computation of conservative gradients are provided in exactly the same form in Bolte and Pauwels 2020a, at least in their arxiv version. Given that, the novelty of this part of the paper is the proof of Theorem 2, which can be found in page 15 and mostly follows a very simple induction argument similar to the existing proofs for the smoothed case.
>
> The Baur-Strassen theorem was originally formulated for rational functions. Extensions were made for smooth functions (Griewank and Walther (2008)); nonsmooth results are almost non-existent. Our result extends this to the vast class of semi-algebraic, or more generally definable functions encompassing virtually all ML losses [15,17]. We do not see this as a simple/minor extension.
>
> The cornerstone of this analysis is the calculus model described in Section 3.1:  an extension of the notion of arithmetic circuit complexity (for polynomials) to nonsmooth functions. This is a substantial improvement which subsumes a sharp knowledge of automatic differentiation, o-minimal structures, and conservative calculus. The fact that further proofs follow simple arguments is precisely because the model is well thought out and consistent. Such a model encompassing  such a wide range of applications is totally absent from the literature (to our knowledge). To conclude on this result and to convey the breadth of its scope, we would like to insist that it allows covering almost all nonsmooth practical problems  we are aware of (through semi-algebraic and o-minimal functions) -- the key being the possibility of Whitney stratification!
>
>
> Finally, a large part of the paper goes beyond conservative calculus, we establish lower bounds on the computational hardness of non-smooth AD and various generalized gradient methods.
>
>
> ### I am not sure I understand the motivation of Theorem 3. Why would we need to compute the directional derivatives since we can compute the whole subgradient?
>
> The relation between subdifferential (the set-valued map), subgradients (elements of the subdifferential), and directional derivative is not simple. For example consider $f(t) = |t|$ and $g = -f$, both subdifferentials at 0 are the same: $[-1,1]$, but the directional derivatives are opposite: $f'(0,d) = - g'(0,d)$. In any case, we argue in Theorem 4 that computing the whole subdifferential is hard (enumerating subgradients is hard).
> Furthermore, given  one subgradient (one single element of the subdifferential), for which there exist oracles, it is not obvious (not possible?) to obtain relevant directional derivative information.
>
> On the other hand, directional derivatives satisfy an exact chain rule (contrary to subdifferential and subgradients). Based on this observation, Khan and Barton used the lexicographic derivatives of Nesterov to develop automatic differentiation algorithms to evaluate elements of $\partial^{c} F$. All these procedures require to evaluate $p$ directional derivatives, and this is why we investigated complexity bounds for such oracles.

---

> > ### Author Response · Authors · 2022-11-10
> > **Response to Reviewer 9fsl (2/2)**
> >
> > ### Related to Theorem 4: The problem of given a function $f$, a vector $v$, and a point $x$ verify whether $v$ is a subgradient of $f$ at $x$ is solvable in polynomial time or not?
> >
> > We do not know the answer to this question. It is usually not investigated in these terms in the literature. As mentioned above, there are algorithmic differentiation routines for Clarke subgradients, so we what we can say is ``computing a subgradient $v$ of $f$ at $x$ can be done in polynomial time''. We believe that the decision version of this problem described by the reviewer should be tractable but currently have no proof
> >
> > ### If the answer to question 3 is positive then I am puzzled with Theorem 4 (b). The authors state that deciding the differentiability of a non-smooth function at a point $x$ is NP-hard. But I believe that the correct statement is that deciding the non-differentiability of $f$ is NP-hard. In fact, if the answer to 3 is positive, deciding the non-differentiability should be NP-complete, since the existence of two subgradients is a witness of the non-differentiability of $f$ at $x$. This means that deciding the differentiability is coNP-complete and we do not expect it to either be in NP or being NP-hard.
> >
> > The reviewer is perfectly right. This is a mistake from our side: the correct statement is indeed the one proposed by the reviewer. Corollary 2 (Appendix page 25) actually contains the correct version (with a wrong title), and we did not report it correctly in the main text. Thank you for catching this; we will correct it.
> >
> > ### There is a small issue with Theorem 4 that usually is swept under the rug, but when proving computational hardness results it needs attention. What is the representation of the numbers that you assume in Theorem 4?
> >
> > (a) If you use binary representation, then the hardness result of Theorem 4 implies that there is no algorithm with complexity polynomial in $p$ (the dimension) and $log(\frac{1}{\epsilon})$, where $\epsilon$ is the accuracy that we choose to represent the numbers.
> >
> > (b) If you can show the hardness result even when you use unary representation, then Theorem 4 implies that there is no algorithm with complexity polynomial in $p$ and $\frac{1}{\epsilon}$.
> >
> > If the case is (a), then this would be a less interesting result because the ML community focuses on algorithms with running time
> > poly($\frac{1}{\epsilon}$). In any case, a discussion about this is missing and is important.
> >
> > Thank you again for this interesting and informative question. Theorem 5 in the appendix contains a hardness result for relu programs with numerical inputs constrained to be in $\{-1,0,1\}$. Therefore, our reduction works using unary representation (there is actually no issue of integer size, only matrix or network size). We believe that the remark is very relevant and decided to hide these details in the appendix to simplify the message.
> > Following the reviewer's comments, we will add a sentence after Theorem 4 explaining that the numerical parameters and inputs are actually constrained to be in $\{-1,0,1\}$.

---

> > > ### Comment · Reviewer_9fsL · 2022-11-17
> > > **Thank you for the clarifications**
> > >
> > > So it seems that it is not even clear whether non-differentiability is inside NP which seems an interesting open problem.
> > >
> > > After reviewers rebuttal I will raise my score to 6.

---

### Official Review · Reviewer_LBds · 2022-11-04

**Confidence:** 4
**Clarity, Quality, Novelty And Reproducibility:** The work is clear, novel and original.
**Correctness:** 4
**Technical Novelty And Significance:** 4
**Empirical Novelty And Significance:** Not applicable
**Recommendation:** 8

**Strength And Weaknesses:**

Strengths:
- Nonsmooth operations/layers are ubiquitous in model neural networks (ReLU activation, sorting operations, max pooling, etc), and backpropagation is the default algorithm to train these networks. Having a rigorous theory of what backpropagation actually computes, and at which cost, is of paramount importance.
- The concept of conservative gradients fills this need, and thus studying it in more detail is important for our understanding of theory and practice.
- The paper is well-written, with a wealth of examples, a motivated impact.

Weaknesses:
- no important ones (see minor clarifications below)

### Minor remarks and question
- In Theorem 2, ii, an upper bound on the ratio is derived where $\omega_f$ depends on the dimension $p$. For completeness, one may add that this upper bound is reached, for example in the case where $|pr(i)|$, and the costs of $g_i$ and $gd_i$ are independent of $i$. Otherwise it could just be a bad upper bound.
- In the first paragraph of Section 5.1: is it known than the matrix multiplication exponent $\omega$ is $> 2$? If $\omega$ were to be equal to $2$, then the overhead ratio of order $p^{\omega - 2 + o(1)}$ could in fact be dimensionless. This is later discussed in the "Consequences" part (and at the end of Section A.2.2) but it could be mentioned earlier: in all rigor, there is currently no result that says $\omega > 2$, and so the derived ratio could be dimensionless (though all practical evidences are against this so far), which contradicts "in contrast with the smooth case with essentially dimensionless overhead.".
- The ReLU requires only 2 bits of encoding > is it not a single bit (a binary value encoding the sign of the input)?


### Very minor cosmetic:
- the authors may consider configuring hyperref through \hypersetup in order to avoid the flashy green and red boxes (see 2nd answer here: https://tex.stackexchange.com/questions/823/remove-ugly-borders-around-clickable-cross-references-and-hyperlinks)
- when the citation is part of the sentence, it should not use parentheses: "see Griewank et al (1989)". When it is not part of the citation, it should use parentheses: "It is at the core of modern learning architectures (Rumelhart et al., 1986).". Use \citet and \citep respectively.
- , little is known about the nonsmooth case. > "but" little is known about the nonsmooth case?
- Standard computational practice of AD consists of: "consists of" means "is composed of". You mean "consists in"
- in an ML > in a ML
- In particular,they > missing space after comma
- To conculde
- Let $F, G$ be locally Lipschitz continuous mapping > mappings
- Lipchitz
- Path differentiable functions are ubiquituous > ubiquitous
- consist of combinations outputs of > consists of combinations *of* outputs of
- is an optimization algorithms > algorithm
- The function F may describe, for instance, a ReLU feedforward neural network empirical loss, parameterized by $p$ real weights, with $q$ inputs: isn't $p$ the number of inputs and $q$ the number of parameters in the previous sentence?
- then evaluate $p$ scalar product > products
- for example $F$ is feedforward neural network > $F$ is a feedforward
- Case 13: This can be done by sorting the n numbers and output the value value > and outputting + "value" duplicated
- Therefore, the each affine part has zero derivative > remove "the"
- fintiely

**Summary Of The Paper:**

The paper studies the computational cost of forward and backward automatic differentiation (AD° for *nonsmooth* programs, based on the notion of conservative gradient introduced by Bolte and Pauwels (2020a, b).
The relative cost of backward mode AD/backpropagation (compared to the original program/function) is proved to be independent of the dimension for path-differentiable functions (that encompass semialgebraic and definable locally Lipschitz functions). Theorem 2 is a nice result that extends results that were known in the smooth case under the name of "cheap gradient principle" (Baur and Strassen, 1983).
Lastly, the authors show that such results on conservative gradients do not hold for other nonsmooth alternatives: the cost ratio for computing $p$ directional derivatives depends on the dimension $p$ unless the matrix exponent is equal to 2 (Theorem 3), and finding two distinct Clarke subgradients in the Clarke subdifferential is NP hard (Theorem 4, by reduction to 3-SAT).

**Summary Of The Review:**

Well-written and theoretical sound paper on important recent development in automatic differentiation for non smooth functions, in particular neural networks.

---

> ### Author Response · Authors · 2022-11-10
> **Response to Reviewer LBds**
>
> We thank the reviewer for his positive evaluation, comments and constructive feedback, we provide a detailed answer to his remarks below.
>
>
>
>
> ### In Theorem 2, ii, an upper bound on the ratio is derived where $\omega_{f}$ depends on the dimension $p$. For completeness, one may add that this upper bound is reached, for example in the case where $pr(i)$, and the costs of $g_i$ and $gd_{i}$ are independent of $i$ . Otherwise it could just be a bad upper bound.
>
> True, the upper bound in ii) is reached, but it is still dependent on the dimension $p$; we will add a remark as suggested.
>
>
>
> ### In the first paragraph of Section 5.1: is it known than the matrix multiplication exponent $\omega > 2$ ? If $\omega$ were to be equal to $2$, then the overhead ratio of order $p^{\omega -2 + o(1)}$ could in fact be dimensionless. This is later discussed in the "Consequences" part (and at the end of Section A.2.2) but it could be mentioned earlier: in all rigor, there is currently no result that says $\omega > 2$, and so the derived ratio could be dimensionless (though all practical evidences are against this so far), which contradicts "in contrast with the smooth case with essentially dimensionless overhead".
>
>
>
> The referee is right. We will add more precision regarding the overhead at the beginning of the section: the actual value of $\omega$ is currently unknown, the best known lower bound is 2.37, and in practice, it is closer to 2.7, both corresponding to a dimension-dependent overhead.
>
> ### The ReLU requires only 2 bits of encoding > is it not a single bit (a binary value encoding the sign of the input)?
>
> Yes, the referee is right. We will change it in the new version. Thanks for catching this.
>
> ### Minor cosmetics
>
> Thanks for pointing these out, we will implement all these suggestions.

---

> > ### Comment · Reviewer_LBds · 2022-11-18
> > **Response to rebuttal**
> >
> > I thank the authors for answering my questions. I am satisfied by their answer.
> >
> > Contrary to Reviewer 9fsL, I think the contribution is sufficient for ICLR: I believe the paper improves our knowledge about conservative calculus, but contributions of theorems 3 and 4 on Clarke subgradients and directional derivatives in the nonsmooth case are also of interest.

---

### Decision · Program_Chairs · 2023-01-20

**Decision:**

Accept: notable-top-25%

**Justification For Why Not Higher Score:**

Although this is an interesting area, and as argued above, nonsmooth functions abound in modern ML, it remains the case that the implications are largely theoretical.

**Justification For Why Not Lower Score:**

A spotlight feels appropriate because although the practical implications are probably sufficiently small that most ML practitioners do not "need" to know this work, it is also the case that many ought to be interested in its existence.

**Metareview: Summary, Strengths And Weaknesses:**

This paper extends the emerging theory of automatic differentiation in the presence of nonsmooth functions.  Given the plethora of nonsmooth operations (Relu, top k, max) in modern ML models, this is a valuable endeavour.

One reviewer expresses concern about the magnitude of the contribution over previous work by Bolte and Pauwels, which the rebuttal does not answer directly, speaking instead to extensions over Griewank and Walther 2008.  On inspection, the arguments show sufficient novelty in the new work.



**Note From Pc:**

if the above contains the word "oral" or "spotlight" please see: "oral" presentation means -> notable-top-5% and "spotlight" means -> notable-top-25%. As stated in our emails, we are disassociating presentation type from AC recommendations